# Surface modification using heptafluorobutyric acid to produce highly stable Li metal anodes

Yuxiang Xie[1], Yixin Huang[1], Yinggan Zhang[2], Tairui Wu[1], Shishi Liu[1], Miaolan Sun[1], Bruce Lee[3], Zhen Lin[3], Hui Chen[1], Peng Dai[1], Zheng Huang[1], Jian Yang[1], Chenguang Shi[1], Deyin Wu[1], Ling Huang[1] ✉, Yingjie Hua[4], Chongtai Wang[4] ✉ & Shigang Sun[1] ✉

The Li metal is an ideal anode material owing to its high theoretical specific capacity and low electrode potential. However, its high reactivity and dendritic growth in carbonate-based electrolytes limit its application. To address these issues, we propose a novel surface modification technique using heptafluorobutyric acid. In-situ spontaneous reaction between Li and the organic acid generates a lithiophilic interface of lithium heptafluorobutyrate for dendrite-free uniform Li deposition, which significantly improves the cycle stability (Li/Li symmetric cells >1200 h at 1.0 mA cm$^{-2}$) and Coulombic efficiency (>99.3%) in conventional carbonate-based electrolytes. This lithiophilic interface also enables full batteries to achieve 83.2% capacity retention over 300 cycles under realistic testing condition. Lithium heptafluorobutyrate interface acts as an electrical bridge for uniform lithium-ion flux between Li anode and plating Li, which minimizes the occurrence of tortuous lithium dendrites and lowers interface impedance.

Li metal has been recognized as one of the most suitable anode materials for electrochemical energy storage systems owing to its high theoretical specific capacity (3860 mAh g$^{-1}$) and low negative electrochemical potential (−3.040 vs. SHE)[1,2]. In particular, Li metal anodes paired with Ni-rich LiNi$_x$Mn$_y$Co$_{1-x-y}$O$_2$ layered cathode materials, such as LiNi$_{0.8}$Co$_{0.1}$Mn$_{0.1}$O$_2$ (NMC811), are among the most efficient material combinations for rechargeable batteries[3]. However, it is highly reactive with most liquid organic electrolytes, particularly with carbonate-based solutions that are very compatible with the currently available 4 V Li-ion layered cathode materials. In carbonate based-electrolytes, Li dendrites growth is often encountered during battery operation, which severely limits the practical applications of Li metal batteries (LMBs).

To prevent the occurrence of side reactions and maintain the interfacial stability of Li metal anode, a stable solid electrolyte interface (SEI) between the Li metal and electrolyte must be formed. In recent years, most of the effective strategies employed to enhance the interfacial stability of Li metal anodes focused on electrolyte design, which include the use of ether-based[4], high-concentration[5], and ionic liquid electrolytes[6], and some electrolyte additives[7]. Therein, a stable SEI layer is introduced either by tuning the electrolyte degradation process or by consuming specific components of the electrolyte. Notwithstanding, parasitic reactions occur at Li metal anode surface when carbonate-based electrolytes are used. Moreover, passivation layers form on the Li surface even under storage in a low contamination glove box due

[1]College of Chemistry and Chemical Engineering, State Key Laboratory of Physical Chemistry of Solid Surfaces, Xiamen University, 361005 Xiamen, China. [2]College of Materials, Xiamen University, Xiamen Key Laboratory of Electronic Ceramic Materials and Devices, 361005 Xiamen, China. [3]Reliability Safety Department & Mechanism Simulation, Contemporary Amperex Technology Co., Limited., 352100 Ningde, China. [4]Hainan Normal University, Key Laboratory of Electrochemical Energy Storage and Energy Conversion of Hainan Province, 571158 Haikou, China. ✉e-mail: huangl@xmu.edu.cn; wangct@hainnu.edu.cn; sgsun@xmu.edu.cn

to its high reactivity[8]. These passivation layers cannot be removed from the Li surface even with the most efficient electrolyte design. During spontaneous chemical reactions, the chemical composition of the passivation layer along the Li surface varies. Consequently, the electrochemical kinetics also changes at different positions along the metal anode, yielding a non-uniform Li-ion flux and triggering the growth of Li dendrites[9]. Iodic acid has been proposed and successfully removes the passivation layer by spontaneous reaction[10]. However, the shuttle effect of iodide ions hinders its extensive application in LMBs. The irreversible self-discharge behavior during the charging process occurs and continuously consume the Li reservoir if paired with high voltage cathode materials[11,12].

In this study, we propose a strategy to rebuild the interfacial layer on the Li surface and simultaneously construct a protective layer with lithiophilic properties (Fig. 1). To achieve this, we treated Li metal using fluorinated carboxylic acid and evaluated the effect of carbon chain length on the cycling stability of Li metal. The results indicated that lithium heptafluorobutyrate (carbon chain length = 4) provided the best protection for the Li anodes. The heptafluorobutyrate acid (HFA)-treated Li anode is denoted as HFA-Li. HFA-Li exhibited excellent lithiophilic properties. After HFA treatment, HFA-Li has enhanced wettability with the electrolyte, which eliminates the passivation layer, increases the Li-ion flux, and lowers the interface impedance, leading to lower lithium deposition overpotential[8,13,14]. Simultaneously, the interfacial layer of HFA-Li effectively homogenizes the Li-ion flux, which in turn induces uniform Li deposition and reduces the formation of isolated Li metal trapped by the SEI[15,16]. Consequently, HFA-Li achieves high Coulombic efficiency (CE) and suppresses the formation of Li dendrites. The cells assembled using HFA-Li exhibited considerably improved stabilities. In particular, the cycle time of the Li/Li symmetric cell increased from 200 to 1200 cycles when the anode was changed from Bare-Li to HFA-Li. Furthermore, highly reversible Li plating/striping processes in carbonate-based electrolytes with a CE as high as 99.3% were realized using the HFA-Li anode. Lastly, the Li||NMC811 full cell exhibited a stable cycle with an 83.2% capacity retention over 300 cycles even under high area cathode loading (20 mg cm$^{-2}$) and limited Li excess (50-μm thick Li foil) conditions. Herein, this study develops a simple surface treatment method to form a lithiophilic interface between the carbonate-based electrolyte and Li that improves the stability of the anode. Therefore, the proposed surface modification strategy can also enhance the stability of LMBs and consequently extend their practical applications.

## Results

### Design mechanism of the surface modification using fluorinated carboxylic acids

Fluorinated carboxylic acid reacts with Li through a spontaneous chemical reaction as illustrated in Fig. 1. In this reaction, the reactive carboxyl group and Li act as the proton and electron donors, respectively. Fluorinated carboxylic acid reacts rapidly with the reducing Li surface through a substitution reaction, thereby forming lithium fluorinated carboxylate and $H_2$ gas. The Li surface contains a passivation layer mainly composed of $Li_2CO_3$ and LiOH, which are formed during its storage in a glove box[8]. Carboxylic acid is stronger than carbonic acid; thus, fluorinated carboxylic acid can react with $Li_2CO_3$ and even with the strong base LiOH and basic $Li_2O$ on the Li surface according to the reactions shown in the supplementary information. In-depth experimental and theoretical calculations are provided to illustrate the removal of the passivation layer (Supplementary Note S1–S3 and Supplementary Figs. S1–S3). Considering these reactions, fluorinated carboxylic acid can be used to remove the passivation layers on the Li surface. At the same time, a lithium fluorocarbon-containing surface layer may be produced through the surface treatment using fluorinated carboxylic acid to improve the cycling stability of Li metal anodes.

Initially, the effect of the carbon chain length of the fluorinated carboxylic acids on the stability of the Li anode was evaluated. Supplementary Fig. S4 shows the voltage–time curves of the Li/Li symmetric cells containing Li anodes treated with fluorinated carboxylic acids of different carbon chain lengths. The cells containing treated Li anodes demonstrated considerably improved cyclic stabilities. Surface treatment using heptafluorobutyric acid (HFA) with a carbon chain length of 4 had the best protection effect on the Li anode. The Li/Li symmetric cell composed of HFA-Li electrodes remained stable even after over 350 times of the plating/stripping processes. On one hand, carboxylate species with short carbon chains, which are similar to the naturally formed lithium carboxylates, such as HCOOLi in methyl formate and $CH_3(CH_2)_2COOLi$ in γ-butyrolactone, in the SEI layers, have limited protective effect on the Li anode[15]. On the other hand, long-chain lithium carboxylates are less flexible due to their long alkyl chains[17,18]. The effect of the C–F functional groups on the stability of Li was also examined. For this purpose, Li anodes were also treated using butyric acid (BA) with a carbon chain length of 4. Unlike HFA, BA contains no fluorine substituents. The performance of the Li/Li symmetric cell containing HFA-Li electrodes was still superior to that of the cell with the BA-Li electrodes (Supplementary Fig. S5). C–F functional groups possibly improved the cycling stability of Li metal. The mechanism of stability improvement corresponding to the presence of

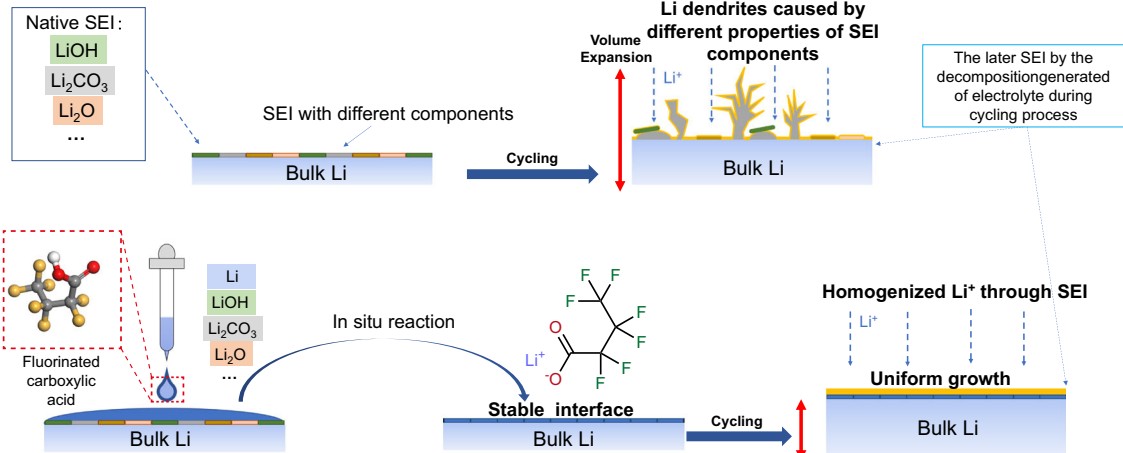

**Fig. 1 | Schematic of lithium metal anodes with and without HFA treatment.** Schematic illustration of the in-situ formation process and protection mechanism of the lithium fluorocarboxylate layer.

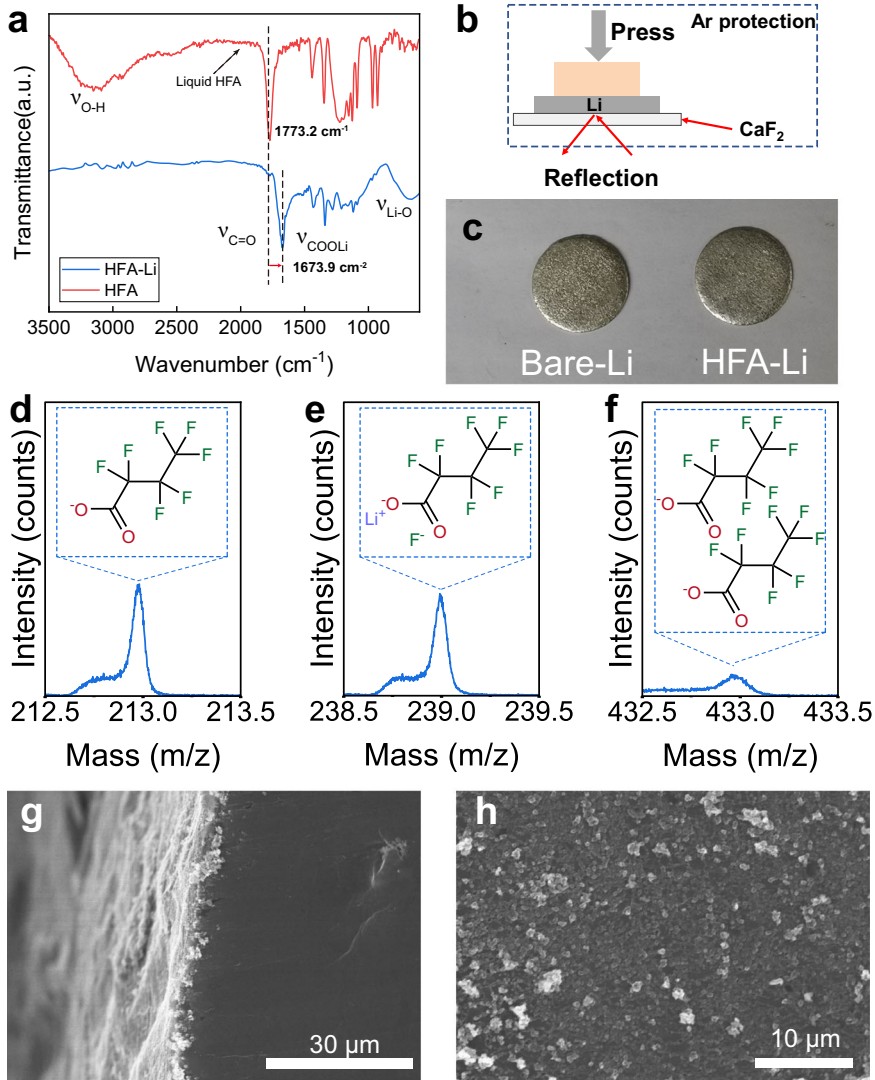

**Fig. 2 | Characterization of the lithium carboxylate protective interface.**
**a** Infrared spectra of HFA-Li and HFA; **b** Schematic of the HFA-Li infrared test setup; **c** Optical images of the HFA-Li and Bare-Li electrodes; **d**–**f** Time-of-flight secondary ion mass spectrometry (TOF-SIMS) of HFA-Li; **g**, **h** Front and cross-sectional SEM images of HFA-Li.

C−F groups will be discussed in more detail in the section on the theoretical calculations. Considering these results, HFA-Li was used in the subsequent tests and characterizations.

**Characterization of the lithium carboxylate protective interface**
Infrared (IR) spectroscopy was performed to verify the formation of the lithium carboxylate protective interface on the Li surface (Fig. 2a). The IR signals of the Li surface were collected through a diffuse reflection mode (Fig. 2b). The peak corresponding to C = O vibration shifted from 1773.2 cm$^{-1}$ (carboxylic acid) to 1673.9 cm$^{-1}$ (HFA-Li), indicating the formation of a metal carboxylate[15]. Moreover, the characteristic broad −COOH peak in carboxylic acid at approximately 2500–3300 cm$^{-1}$ disappeared after the reaction, which indicates the substitution of the −H atoms with −Li atoms[19]. Furthermore, the Li−O peak at 550 cm$^{-1}$ observed at the IR spectrum of HFA-Li is indicative of the formation of the Li−O bonds through the reaction between Li and carboxylic acid. Figure 2c shows the optical images of Li metal anodes before and after the treatment. No apparent change in the color and metallic luster of the Li surface was observed after the treatment with HFA possibly because the formed interface was relatively thin. Time-of-flight secondary ion mass spectrometry (TOF-SIMS) was performed to

study the chemical composition of HFA-Li (Fig. 2d–f and Supplementary Fig. S6). Ion fragments of $(CF_3-CF_2-CF_2)^-$, $(CF_3-CF_2-CF_2COO)^-$, $(CF_3-CF_2-CF_2COOLi)F^-$, and $(CF_3-CF_2-CF_2COO)_2Li^-$ appeared in the spectrum with corresponding mass-to-charge ratios (m/z) of 169, 213, 239, and 433, respectively. The appearance of these ionic fragments corresponds well to the formation of lithium fluorocarboxylate, indicating that the reaction proceeds on the Li surface as described in Fig. 1. Scanning electron microscopy (SEM) was performed to study the morphology of the protective interface on the HFA-Li surface (Fig. 2g, h, and Supplementary Fig. S7). The formation of the protective interface can be observed on the top and cross section views of the HFA-Li electrode. In contrast to flat polymeric coatings, the surface layer was composed of agglomerated small particles. The thickness of the coating was approximately 2 μm (Supplementary Fig. S7).

X-ray photoelectron spectroscopy (XPS) was carried out to investigate the changes in the surface composition of the Li anode after the treatment with HFA (Supplementary Fig. S8). No XPS signal corresponding to C−F was observed at the F 1s spectrum of Bare-Li. In contrast, a C−F signal was observed on that of HFA-Li at approximately 688.7 eV[20]. At the C 1s spectrum of Bare-Li, peaks at 284.8, 287.1, and 289.8 eV were recorded, which correspond to C−C, C−O, and

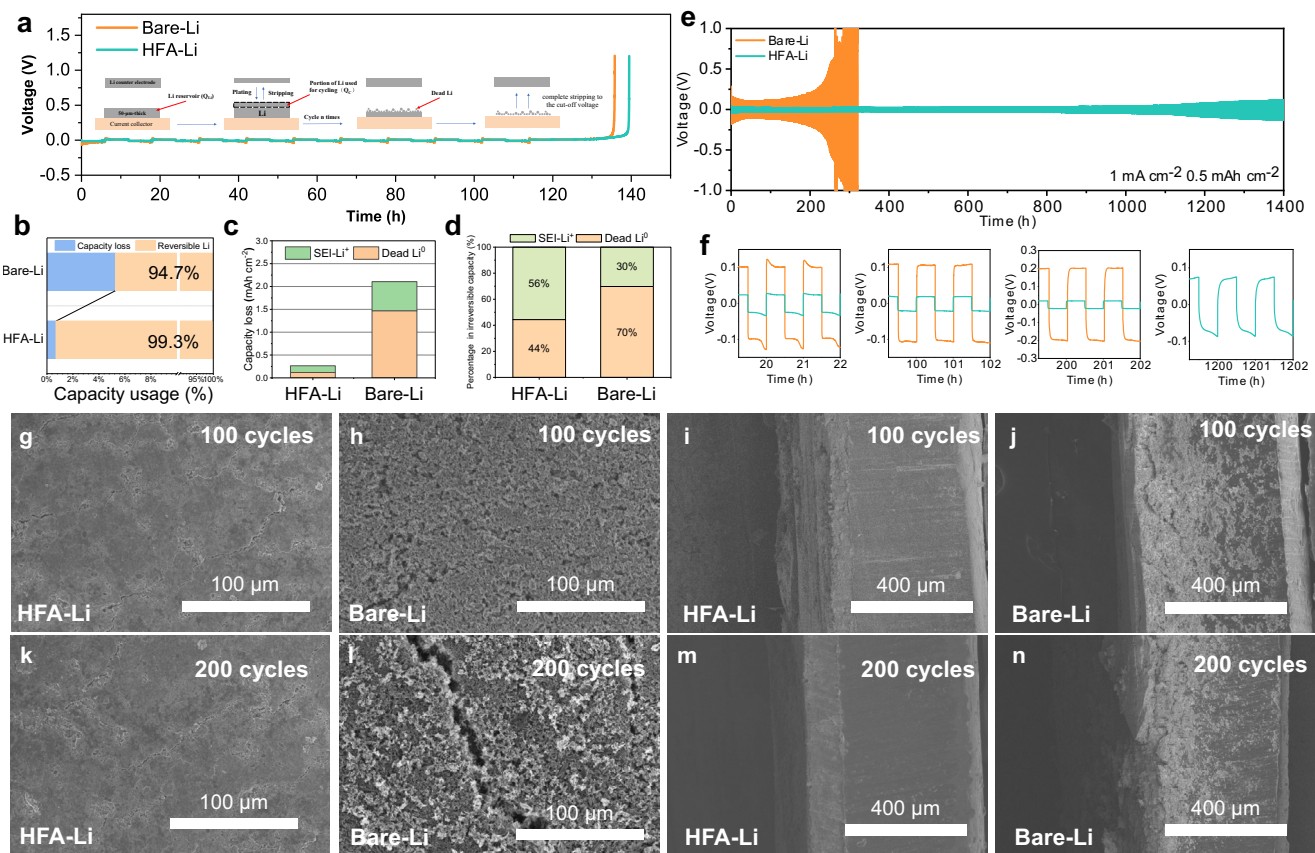

**Fig. 3 | Effect of HFA-Li on Li dendrite growth suppression. a** Voltage profiles of the HFA-Li and Bare-Li anodes during the Coulomb efficiency test in a 1 M LiPF$_6$ in EC/EMC (v/v = 3:7) electrolyte with 5.0 wt% FEC; **b** Analysis of capacity usage (capacity loss and reversible Li) and **c**, **d** capacity loss (SEI Li$^+$ and unreacted metallic Li$^0$) in CE test by the MST method; **e** Voltage–time curve of the Li/Li symmetric cells in a 1 M LiPF$_6$ in EC/EMC (v/v = 3:7) electrolyte with 5.0 wt% FEC at a current density of 1.0 mA cm$^{-2}$ and a capacity of 0.5 mAh cm$^{-2}$; **f** Corresponding magnified voltage–time curves of the Li/Li symmetric cells; **g**–**n** Front and cross-sectional SEM images of the Li/Li symmetric cells containing Bare-Li and HFA-Li after 100 and 200 cycles of plating/striping processes;.

carbonates (O−C=O), respectively. After the HFA treatment, new peaks at 288.3, 291.2, and 293.6 eV were observed at the C 1*s* spectrum, which can be attributed to C=O, C−F, and C−F$_3$, respectively. The intensity of the peaks corresponding to the C−O and C=O groups increased after the treatment. These results further confirm the successful formation of a lithium carboxylate protective layer on the Li surface after HFA treatment.

### Effect of HFA-Li on suppressing Li dendrite growth

The Coulomb efficiency (CE) of the Li striping/plating process is a critical parameter of the cycle performance of battery systems. Since the lithiophilic protective interface of the HFA-Li anode was formed in situ on the Li surface, the CE cannot be determined through the Li/Cu cell method. In this study, an improved test method using thin Li foil was employed, which is closer to the practical cycle conditions of the LMBs. Figure 3a shows the measured voltage profiles of the HFA-Li and Bare-Li anodes in a 1 M LiPF$_6$ in EC/EMC (v/v = 3:7) electrolyte with 5.0 wt% FEC. The schematic of the plating/stripping processes is also presented. The average value of the CE of Bare-Li was only 94.7% (Fig. 3b). This implies a large irreversible capacity loss possibly due to the generation of Li dendrites during cycling, which eventually became dead Li after Li striping[21]. In contrast, the HFA-Li anode registered a much higher CE (99.3%), which further confirms the uniform Li$^+$ ion flux and Li deposition, and inhibited dendritic Li growth during cycling of HFA-Li. To support the above and to gain more insight into the origin of the capacity loss, mass spectrometry titration (MST) technique was further performed to distinguish the contribution of dead Li$^0$ (metallic dead Li metal wrapped by SEI) and SEI-Li$^+$ (SEI components) in

the CE test (Fig. 3c, d)[22–24]. The accumulated dead Li$^0$ and SEI-Li$^+$ of HFA-Li during CE test were 0.118 and 0.148 mAh cm$^{-2}$, respectively, while those of Bare-Li were 1.470 and 0.636 mAh cm$^{-2}$. In particular, the accumulated dead Li$^0$ of Bare-Li occupies 70% of the irreversible capacity loss, demonstrating that metallic dead Li metal is the main source of capacity loss during Li plating/stripping. While after the HFA treatment, SEI-Li$^+$ became the main source of irreversible capacity loss. The percentage of dead Li$^0$ decreased to 44% and the irreversible capacity loss was significantly reduced (from 2.106 to 0.266 mAh cm$^{-2}$). This result indicates that the Li deposition behavior of HFA-Li was effectively regulated after HFA surface treatment, which minimized the curvature of the microstructure and achieved a uniform morphology. Consequently, dead Li$^0$ caused by isolated Li particles trapped in the SEI during Li stripping was significantly reduced[21].

Li/Li symmetric cells were used to evaluate the electrolyte-blocking feature of the HFA-Li coating. Supplementary Fig. S9 shows the impedance evolution of Bare-Li/Bare-Li and HFA-Li/HFA-Li symmetric cells over time in a 1 M LiPF$_6$ in EC/EMC (v/v = 3:7) electrolyte with 5 wt% FEC. The HFA-Li symmetric cell exhibited a low and stable interfacial impedance over the entire cell resting time. In contrast, the impedance of the Li/Li symmetric cell assembled using the Bare-Li electrodes increase sharply after the cell assembling to 32 h (from ~110 Ohm·cm$^2$ to ~190 Ohm·cm$^2$).

The long-term cycle stabilities of Li/Li symmetric cells assembled using the HFA-Li and Bare-Li electrodes were also evaluated. Figure 3e, f shows the voltage–time curves of the Li/Li symmetric cell at a current density of 1.0 mA cm$^{-2}$ and a capacity of 0.5 mAh cm$^{-2}$ in 1 M LiPF$_6$ in EC/EMC (v/v = 3:7) with 5.0 wt% FEC. The HFA-Li symmetric cell

exhibited excellent stability even after 1400 h. In contrast, the cell potential of the Li/Li symmetric cell assembled using the Bare-Li electrodes began to fluctuate during the early stages of the stability testing and increased sharply after 200 h only. The above phenomenon can be explained by the following reasons: On one hand, as previously discussed, the Li deposition rate on the Li surface varies at different points on the substrate due to the inhomogeneous composition of the inherent SEI layer[25]. During the Li plating process, the internal Li dendrites continuously grew on the Bare-Li surface. On the other hand, during the Li stripping process, the Li dendrites generated large amounts of dead Li species, which consumes the limited electrolyte and active Li, and hinders the subsequent Li ion transport due to the spatial obstruction[4]. Consequently, this gradually increases the impedance and, eventually, the overpotential. In contrast HFA-Li exhibited a lower overpotential than Bare-Li during cycling. And the cell potential of the HFA-Li symmetric cell in the carbonate electrolyte was stable, indicating that the growth of the Li dendrites during cycling was effectively suppressed. In particular, the performance of the HFA-Li symmetric cell fabricated in this study is better than those of previously reported Li/Li symmetric cells in carbonate- and ether-based electrolytes (Supplementary Table S1).

Figure 3g–n shows the SEM images of the Li/Li symmetrical cells containing HFA-Li and Bare-Li electrodes after 100 and 200 cycles. The formation of Li dendrites on the Bare-Li surface was apparent. The dendrites exhibited a loose porous structure, which is indicative of the non-uniform deposition of Li. In contrast, the HFA-Li surface was very flat and compact. From the cross-sectional SEM image of HFA-Li, the deposited Li layer was uniform, further confirming that the Li fluorinated carboxylate interface promotes uniform Li deposition and effectively inhibits Li dendrite formation.

Fluoroethylene carbonate (FEC) is an effective SEI-forming additive that can enhance the stability of Li metal anodes[26]. Herein, the performance of the HFA-Li anode in a carbonate electrolyte without FEC was also investigated. Supplementary Fig. S10 shows the voltage–time curves of the Li/Li symmetric cell in a 1 M LiPF$_6$ in EC/EMC (v/v = 3:7) electrolyte. At 1.0 mA cm$^{-2}$ and 0.5mAh cm$^{-2}$, HFA-Li exhibited 350 h of cycling. At 2.0 mA cm$^{-2}$ and 1.0 mAh cm$^{-2}$, HFA-Li can be cycled 220 h (Supplementary Fig. S11). In contrast, the Bare Li anode can be cycled for only 120 and 40 h, respectively. In addition, HFA-Li registered a CE toward the Li plating/stripping process as high as 93.71% (Supplementary Fig. S12). The value of the CE of the Bare-Li anode was only 78.93%. The above results indicated that HFA-Li still exhibits better electrochemical performance in carbonate electrolytes without FEC. And the addition of FEC in the electrolyte considerably improved the performance of the HFA-Li anode. As for Bare-Li, when FEC was introduced into the electrolytes, despite the subsequent formation of a LiF-rich SEI layer, the limitations imposed by the inner SEI layer compromise the stability of Li metal anode[25,26]. After HFA treatment, the native SEI layer is removed from the HFA-Li surface, thereby forming a uniform lithium fluorocarboxylate containing layer. The LiF-rich SEI layer formed on the surface during the subsequent FEC cycling decomposition can further enhance the protection effect on Li metal anode as in the mechanism shown in Supplementary Fig. S13.

## Electrochemical performance of the Li||NMC811 full cells
Figure 4a–c shows the full cell test results recorded using a LiNi$_{0.8}$Co$_{0.1}$Mn$_{0.1}$O$_2$ cathode (NMC811) with a high area loading of 20 mg cm$^{-2}$ and 50-μm thick Li foil anode. The HFA-Li || NMC811 full cell demonstrated a stable cycle of over 300 cycles with 83.2% capacity retentions. Although the electrolyte-induced resistive SEI built up during Li plating/striping, leading to an increase in polarization, especially in the early stage of battery cycling, HFA-Li homogenized the Li ion flux, which in turn induces uniform Li deposition and minimized the occurrence of tortuous lithium dendrites. As a result, no considerable capacity drop and cell sudden death were observed

during cycling. In contrast, the unstable interface between Bare-Li and electrolyte accelerated the exposure and growth of fresh lithium dendrites, leading to the fast depletion of Li reservoir. The capacity of the Bare-Li||NMC811 full cell rapidly decayed after only 50 cycles. The continuous formation of dead Li in the Bare-Li anode and continuous consumption of the limited electrolyte during the cycling severely affected the ion transport, which led to the abrupt reduction in the capacity of the Bare-Li||NMC811 cell. The observed rapid capacity decay is indicative of the instability of the interface between the Bare-Li anode and electrolyte. In particular, the performance of the HFA-Li || NMC811 full cell fabricated in this study is better than those of previously reported Li||NMC811 full cells in carbonate- and ether-based electrolytes (Supplementary Table S2).

Electrochemical impedance spectroscopy (EIS) measurements were also performed to evaluate the change in the electrode/electrolyte interface properties of Bare-Li and HFA-Li after a long-term cycle. Figure 4d–f shows the EIS tests profiles of the Li||NMC811 cells after 2, 50, and 100 cycle of the full cell testing. All spectra consist of a semicircle at high and medium frequencies, respectively, and a diagonal line at low frequencies, which correspond to the charge transfer resistance ($R_{ct}$), the impedance of SEI film ($R_{SEI}$) and Warburg impedance, respectively. In particular, The value of Z′ at the highest frequency is defined as $R_s$, which correspond to interface impedance causing by side reaction and dendrites growth[4,27]. After the initial two cycles, the values of $R_{ct}$, $R_{SEI}$, and $R_S$ for HFA-Li and Bare-Li were similar. As the cycle proceeded to 50 and 100 cycles, the $R_{ct}$, $R_{SEI}$, and $R_S$ of both HFA-Li and Bare-Li gradually increased. However, it can be observed that the values of $R_{ct}$ and $R_{SEI}$ are still close to each other after cycles, while the $R_s$ of Bare-Li are significantly larger than that of HFA-Li, indicating that the cells assembled by Bare-Li undergo more severe internal side reactions and dendrites growth during cycling.

## Lithiophilic protective interface of the HFA-Li anode
To probe the critical effect of HFA-Li on uniform lithium-ion flux and the lithiophilic performance of Li metal anode, the interactions of HFA-Li, BA-Li, and single Li ion with the Li substrate were evaluated through density functional theory (DFT) calculations. Figure 5a, b and Supplementary Fig. S14 shows the stable configurations and corresponding charge density difference of BA-Li, HFA-Li and Li ion on the surface of Li (100). The blue and yellow regions represent charge loss and accumulation, respectively (Fig. 5a, b). The absorption of HFA-Li has a significant impact on the electronic state and charge distribution of the surrounding area, leading to an electron transfer from Li to O atoms surrounding the region[15]. The adsorption energies of HFA-Li and BA-Li on a Li substrate were determined to be −2.11 eV and −1.99 eV, respectively, after fully optimizing the structure. For comparison, the Li adsorption energy of the Li substrate was lower at −1.61 eV (Supplementary Fig. S14). The PDOS (Projected density of state) of adsorbed Li on Li substrate with BA-Li and HFA-Li were also evaluated based on DFT calculations (Fig. 5c, d). When HFA-Li was introduced, the O atom of HFA-Li interacted significantly with the Li adatom compared to the introduction of BA-Li. The high orbital hybridization of Li-s and O-2p states indicates the strong interactions between HFA-Li and adsorbed Li on Li substrate. The adsorption energy of the Li substrate for HFA-Li was also higher than that for BA-Li. This can be attributed to the strong electron-withdrawing effect of the C−F functional groups, which further promotes the electron transfer and enhances the HFA-Li adsorption capacity[28]. And the strong adsorption energy of HFA-Li with Li surface can suppress the longitudinal growth of Li and facilitate the lateral growth of Li to achieve uniform Li deposition[29]. Owing to their strong interaction with the Li substrate, the O atoms in the carboxylic acid group can act as nucleation sites for Li deposition to homogenize Li ion flux, promote uniform Li deposition, and consequently improve the lithiophilic character of the interface[15,16]. Therefore, the carboxyl and C−F functional groups of HFA-Li were possibly

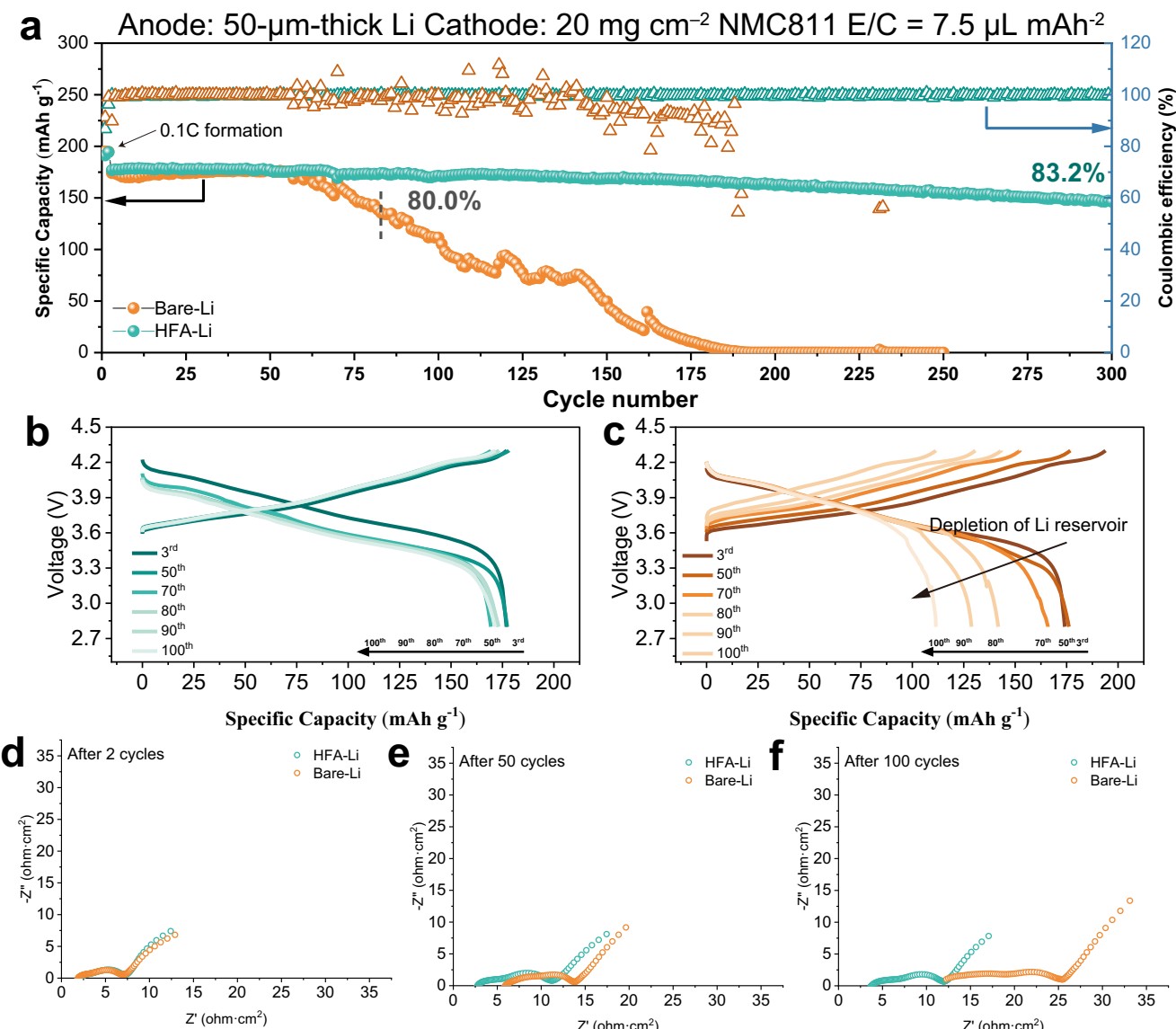

**Fig. 4 | Electrochemical performance of the Li ||NMC811 full cells. a** Long-cycling performance of the Li ||NMC811 cell; Charge–discharge curves of **b** HFA-Li and **c** Bare-Li taken at the 3rd, 50th, 70th, 80th, 90th, and 50th cycle of the full cell testing. Conditions: 50-µm thick Li, high area loading NMC811[4.0 mAh cm⁻², 20 mg cm⁻² (1C = 200 mA g⁻¹)]. The cells were activated at 0.1 C for 2 cycles, then charged at 0.2 C and discharged at 1.0 C in subsequent cycles; **d–f** Electrochemical impedance spectroscopy (EIS) tests of the Li ||NMC811 cells after 2, 50, and 100 cycle of the full cell testing.

critical to ensure uniform Li⁺ ion flux and Li deposition, and suppress Li dendritic growth during cycling.

To investigate the wettability of the Bare-Li and HFA-Li surfaces in carbonate electrolytes, contact angle measurements were performed (Fig. 5e). The contact angle of the Bare-Li surface (26.2°) is much higher than that of HFA-Li (12.5°). HFA-Li exhibited better wettability than Bare-Li possibly due to the presence of polar groups on its surface, which facilitated its interaction with the polar electrolyte[13]. Therefore, HFA treatment improved the affinity of the Li surface with the carbonate-based electrolyte. Higher affinity and wettability promote efficient Li ion transport, which homogenizes the distribution of the Li ions near the Li anode and reduces the internal impedance of the cell[16].

To directly observe the effect of HFA treatment on the Li deposition behavior, in situ electrochemical optical microscopy was performed wherein an optical electrochemical cell made of a poly-tetrafluoroethylene chamber with a quartz window was used (Supplementary Fig. S15). Figure 5f shows the real-time optical images of the in-situ Li deposition on the HFA-Li and Bare-Li substrates at a

current density of 5 mA cm⁻² taken after 0, 2, 4, 6, 8, and 10 min. In addition, videos of the dynamic Li deposition process are also included in the Supporting Information (Supplementary Movies 1 and 2). Li dendritic growth can already be observed after 2 min of plating on the Bare-Li surface. As the deposition process progressed, the dendritic growth intensified, which formed inhomogeneous and large Li dendrites on the Bare-Li surface. Eventually, moss-like regions with unevenly distributed Li dendrites were formed. Due to the variation of the chemical composition of the pristine SEI along the Bare-Li surface, the kinetics of Li deposition vary at different points on the substrate, resulting to the non-uniform flux of Li ions and promoting the growth of the Li dendrites[30]. In contrast, the growth of Li dendrites on the HFA-Li surface was effectively suppressed. Li dendrites were not formed on HFA-Li even after 10 min of deposition. Only a gradual change in the luster of the HFA-Li surface was observed, which can be attributed to the uniform deposition of fine Li grains all over the HFA-Li substrate. As such, Li ions can be deposited uniformly on the Li metal surface after HFA treatment. HFA treatment removes the inherent passivation layer

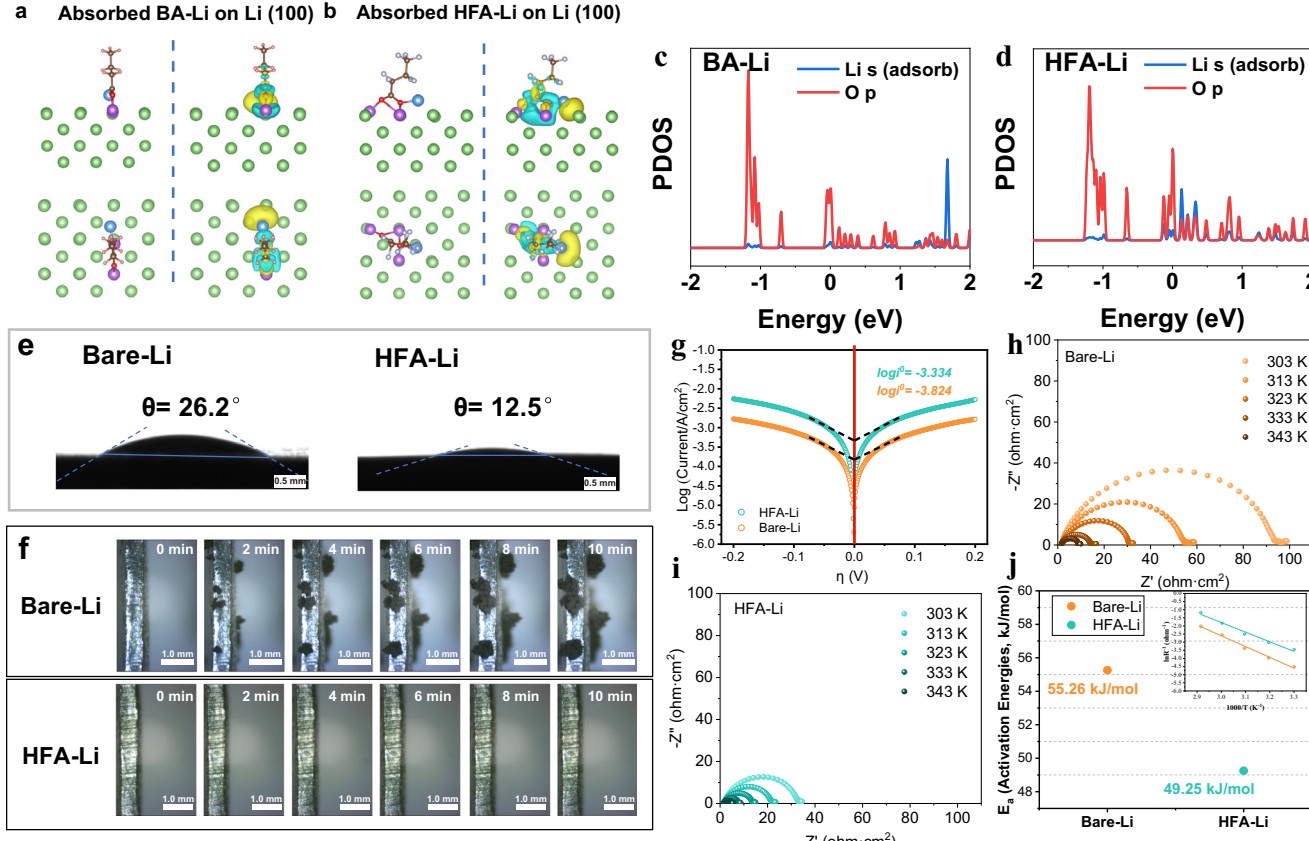

**Fig. 5 | Interface properties of HFA-Li anodes.** Stable configurations and corresponding charge density differences of **a** BA-Li and **b** HFA-Li on Li (100) surface. The brown, red, pink, green, gray, and purple balls represent C, O, H, Li, F, and adsorbed Li atoms, respectively. The yellow and blue regions represent charge accumulation and loss, respectively; The corresponding PDOS of the adsorbed Li atoms and its nearest neighboring O atoms in BA-Li (**c**) and HFA-Li (**d**). **e** Contact angles of Bare-Li and HFA-Li; **f** Optical images of the in-situ Li deposition on Bare-Li and HFA-Li at a deposition current density of 5 mA cm$^{-2}$; **g** Tafel plot of HFA-Li and Bare-Li; EIS plots of the Li/Li symmetric cells containing (**h**) Bare-Li and (**i**) HFA-Li at different temperatures before cycling; (**j**) Activation energy ($E_a$) of HFA-Li and Bare-Li. The inset shows the Arrhenius behavior of the resistant.

on the Li surface while producing a new artificial SEI protective interface. The formation of the lithiophilic interface changed the surface energy of the anode and in turn, the energy landscape of ion deposition, thereby transforming the Li deposition behavior on the Li surface. Figure 5g shows the Tafel plots of the HFA-Li and Bare-Li anodes. Supplementary Fig. S16 shows the corresponding voltage–current curves. The exchange current density ($i^0$) is obtained from the Tafel equation[21]. The value of $i^0$ of HFA-Li (0.463 mA cm$^{-2}$) is higher than that of Bare-Li (0.150 mA cm$^{-2}$), which implies that the charge transfer processes are faster at the electrode/electrolyte interface of HFA-Li than those on Bare-Li.

Electrochemical impedance spectroscopy (EIS) was performed at different temperatures to study the effect of HFA treatment on the charge transfer resistance of HFA-Li and Li deposition activation energies. EIS was also carried out on Li anodes modified using different HFA concentrations (Supplementary Fig. S17). Li/Li symmetric cells were assembled using the HFA-Li and Bare-Li electrodes. EIS measurements were recorded before cycling (Fig. 5h and i). Table S3 and S4 summarize the values of $R_s$ and $R_{ct}$ obtained through equivalent circuit fitting (Supplementary Fig. S18). The diameters of the semicircles in the Nyquist plots of the HFA-Li anode obtained at different temperatures were smaller than those of Bare-Li, indicating that HFA treatment decreased the $R_{ct}$ of Li metal anode. HFA-Li registered a higher value of $i^0$ and lower $R_{ct}$, which indicates that the energy barrier for Li deposition on the HFA-Li surface was lower than that on the Bare-Li substrate. To investigate the mechanism further, the activation energies ($E_a$) for Li deposition on the HFA-Li and Bare-Li surface were calculated[31]. The lower activation energy of the HFA-Li surface (49.25 kJ mol$^{-1}$) than the Bare-Li sample (55.26 kJ mol$^{-1}$) reflects the lower Li deposition energy barrier on HFA-Li and suggests the strong lithiophilic character of the HFA-treated Li anode (Fig. 5j).

## Discussion

In summary, we demonstrate that heptafluorobutyrate can simultaneously remove the surface passivation layer and construct a lithium fluorocarboxylate protective interface on Li surface. Particularly, lithium heptafluorobutyrate with a carbon chain length of four provides the best protection for Li anodes. The protective interface produced after HFA treatment improves the chemical affinity of Li surface with the carbonate-based electrolyte, regulates Li deposition behavior, and promotes uniform Li deposition, thereby enhancing the cycling stability of Li metal anode. Li/Li symmetric cells and Li‖NMC811 full cells assembled using the HFA-Li anode exhibit considerably improved cycle stability compared to those produced using Bare-Li. The route proposed in this work proves to be useful in the design and development of stable Li metal anodes of next-generation LMBs.

## Methods
### Fabrication of the HFA-Li anode

Difluoroacetic acid (DFA, 98%, Aladdin), pentafluoropropionic acid (PFA, 98%, HEOWNS), heptafluorobutyric acid (HFA, 98%, Aladdin), perfluorovaleric acid (PFVA, 98%, Aladdin), undecafluorohexanoic Acid (UFA, 98%), butyric acid (BA, 99%), and tetrahydrofuran (THF, anhydrous, ≥99.9%, inhibitor-free, Aladdin) were stored and used in

glove box ($O_2$ < 0.1 ppm, $H_2O$ < 0.1 ppm) without any purify. The fabrication of HFA-Li anodes was carried out in an argon-filled glove box ($O_2$ < 0.1 ppm, $H_2O$ < 0.1 ppm). Briefly, the HFA solution (0.5 wt%) was prepared by dissolving HFA in anhydrous THF under stirring for 3 h. Subsequently, 100 μL of the above HFA solution was dropped on the surface of a lithium foil (12 mm in diameter) and was placed for 12 h at room temperature to evaporate THF to obtain the HFA-Li anode. DFA-Li, PFA-Li, HFA-Li, PFVA-Li, UFA-Li, and BA-Li were prepared using the same method, except for the different organic acids. The fabrication methods for the HFA-Li electrode in the Li/Li symmetric cell and Li||NMC full cell tests were the same, except that the former used 500 μm thick lithium foil and the latter used 50-μm-thick lithium foil.

## Material characterization

Infrared spectra of HFA-Li and Bare-Li were performed on a Fourier transform infrared spectroscopy (FTIR, IRPrestuge-21, Shimadzu) with wavelength of 400–3500 $cm^{-1}$. Hitachi S-4800 SEM was used to characterize the morphology of Li. Time-of-flight secondary ion mass spectrometry of HFA-Li was conducted on PHI nanoTOF II Time-of-Flight SIMS (30 keV, 2 nA, Ion species: $Bi^{3++}$) Real-time optical images of Li deposition was conducted by an SG900 three-eye stereomicroscope (Suzhou Shenying Optical Instrument Co.). In the in-situ electrochemical optical microscopy experiment, the thickness of the lithium foil was 0.5 mm. The in-situ electrochemical optical cell was designed as shown in Fig. S5. The contact angles of HFA-Li and Bare-Li were determined using a Kruss DSA25E Drop Shape Analyzer. XPS spectra were obtained by a Thermo Fisher Scientific XPS analysis system (Al Kα X-ray source, 1486.8 eV). Mass spectrometry titration (MST) technique was conducted using SHP8400PMS-L online mass spectrometry (Shanghai Sunny Hengping Scientific Instrument Co., Ltd.) with $D_2O$ as the titrator. X-ray diffraction (XRD) analysis was conducted on a Brucker D8 X-ray analyzer, with a scan rate of 2° per minute, using Cu Kα radiation at 40 kV and 40 mA.

## Electrochemical measurements

The high area loading $LiNi_{0.8}Co_{0.1}Mn_{0.1}O_2$ (NCM811) cathode electrodes (20 mg $cm^{-2}$, 4.0 mAh $cm^{-2}$, thickness of 70 μm, and diameter of 12 mm) were provided by CATL (Contemporary Amperex Technology Co., Limited). The active material in the composite cathodes is 95 wt%. Li foils (12 mm in diameter) were purchased from CEL (China energy lithium Co., Limited). Ethylene carbonate (EC), Ethyl methyl carbonate (EMC), fluoroethylene carbonate (FEC), and $LiPF_6$ were purchased from DoDoChem and used as received. Electrolytes preparation were performed in a glove box. Li/Li symmetric and Li/NMC full cells were both assembled in 2032-type coin cells using a Celgard 2025 separator (19 mm in diameter). The electrolyte content of each coin cell was 40 μL. The applied electrolyte was 1 M $LiPF_6$ dissolved in a mixture of EC and EMC in a volume ratio of 3:7 with 5.0 wt% FEC, if not specifically stated. The assembly operations were performed in a glove box filled with argon gas ($O_2$ < 0.1 ppm, $H_2O$ < 0.1 ppm). The cells were tested galvanostatically using a Neware battery testing system (CT-4008). The Li||NCM811cells were activated at 0.1C (current density: 1C = 4.0 mA $cm^{-2}$) for 2 cycles, charged at 0.2C, and discharged at 1.0C in subsequent cycles. And the full cells were cycled between 2.8 and 4.3 V. Tafel curves were obtained by an electrochemical workstation (CHI 660E) in a three-electrode cell using Li as reference electrode and auxiliary electrode. Electrochemical impedance spectroscopy (EIS) were carried out on a Parstat 2263 workstation in the frequency range of 100 KHz to 10 mHz with an amplitude of 5 mV. All electrochemical tests were performed in a thermostatic chamber at 25 °C if not otherwise stated.

To evaluate the CE of the HFA-Li anode in the carbonate electrolyte, a 50-μm-thin Li foil was used as a Li reservoir for the 50 μm-thin Li/Li cells. First, a given amount of charge from the 50 um-thin Li foil ($Q_{Li}$) was used as the Li source. Then, a portion of the charge ($Q_c$) was used

between the Li working and counter electrodes. After $n$ cycles, the remaining Li reservoir was completely stripped to the cut-off voltage. By measuring the amount of Li remaining after cycling, the average CE over $n$ cycles can be calculated by the equation,

$$CE = \frac{nQ_c + Q_s}{nQ_c + Q_{Li}} \quad (1)$$

where $Q_S$ corresponds to the final stripped charge or the amount of Li remaining after $n$ cycles. Herein, the value of $Q_{Li}$ of the 50-μm-thin Li foil was ~10 mAh $cm^{-2}$. Li was cycled at $Q_C$ = 3 mAh $cm^{-2}$ at 0.5 mA $cm^{-2}$ $Q_S$ was measured after stripping the remaining Li at 0.5 for 10 cycles. Finally, mA $cm^{-2}$ to 1.2 V.

The exchange current density was obtained from the Tafel equation

$$\eta = A \times (\log i - \log i^0) \quad (2)$$

where $\eta$ and $A$ represent overpotential and the kinetic constant in the diffusion-controlled surface dynamic range, respectively. And $i^0$ represents the exchange current density. The value of $i^0$ can be obtained from the intersection of the extrapolated linear part of the log $i$ versus $\eta$ plot with the μ = 0 line.

The method to obtain the activation energies of Li deposition was as follows. First, Li/Li symmetric cells were assembled as mentioned above. Then, EIS tests at different temperatures were carried out. The Li deposition activation energy ($E_a$) can be obtained by the following equation:

$$\frac{1}{R_{ct}} = A_0 e^{-E_a/RT} \quad (3)$$

where $R_{ct}$, $A_O$, R and $E_a$ represent the charge-transfer resistance, pre-exponential constant, the standard gas constant and the activation energy, respectively. Therefore, $E_a$ can be extracted from the slope plot of log $R_{ct}$ vs. inverse temperature (1/T)[31].

## Computational methods

First-principles calculations based on density functional theory (DFT) were conducted in the Vienna ab initio simulation package (VASP)[32,33] with the results dealt by the ALKEMIE platform[34]. The Perdew-Burke-Erzenhof (PBE) functional in the generalized gradient approximation (GGA) is used to describe the commutative correlation term in the DFT calculation[35]. The cutoff energy was set as 500 eV for the plane wave basis to ensure the precision of calculations, and the pseudopotential used is projector augmented wave (PAW) pseudopotential[36]. A vacuum layer of 20 Å was introduced in the calculations to avoid the interaction between images. Both the relaxation convergence for ions and electrons were 1 × $10^{-5}$ eV. The k-points of 1 × 1 × 1 were generated with Gamma symmetry automatically. The binding energy ($E_b$)[37] was calculated by the equation

$$E_b = E_{Li/slab} - E_{slab} - E_{Li} \quad (4)$$

where $E_{Li/slab}$ is the total energy of the slab with an adsorbed Li atom, $E_{slab}$ and $E_{Li}$ are the total energies of the slab and a Li atom, respectively. For calculation of reaction free energy, Empirical correction of the Grimme's scheme[38] (DFT-D3) is used to describe van der Waals interaction. VASPsol program is used to take solvation effect into consideration, and the solvent is water whose dielectric constant is 78.4[39]. KPOINTS is generated by VASPKIT program[40]. The K-Mesh is Monkhorst-Pack Scheme[41], and the resolved value is 0.025. The single molecule is in the cube cell with a side length of 25 Å. The Gibbs free energy is obtained by vibration analysis and statistical thermodynamics, and the temperature is 298.15 K.

## Data availability

The data that supports the findings of this manuscript can be found in the Supplementary Information and are available free of charge or from the corresponding author upon request. The raw data of the figures shown in the main manuscript are available from figshare with the identifier https://doi.org/10.6084/m9.figshare.22331410.

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

## Acknowledgements

This work was supported by the National Key Research and Development Program (No. 2021YFB2400300) and Natural Science Foundation of China (grant numbers 22172133, 22288102) and Research Project of Hainan Academician Innovation Platform (grant numbers YSPTZX202038).

## Author contributions

Y.-X, S.-S., L.H., and C.W. conceived and designed the research. Y.-X., P.D., Z.H., B.L., Z.L., J.Y., Y.Z., T.W., D.W. C.-S. performed materials synthesis and characterization and data analysis. Y.-X., Y.H., S.L., M.S., and H.C. conducted battery preparation and electrochemical measurements. Y.-X. and L.H. wrote the manuscript. S.-S., L.H., B.L., Y.H., and C.W. revised the manuscript. All authors were contributing to the discussion of the study.

## Competing interests

The authors declare no competing interests.
