## [Peer Review File · Nature Communications]

Surface modification using heptafluorobutyric acid to produce highly stable Li metal anodesREVIEWER COMMENTS

Reviewer #1 (Remarks to the Author):

This is an article on surface modification of lithium metal (aka artificial solid electrolyte interphase).

Answering this journal's queries:

*There are no noteworthy results in this ms.

*The ms has little significance due to lack of novelty.

*The work does not provide novel, credible, or significant scientific insight.

* The flaws of data analysis etc. are not of concern given the mundaneness of research.

The approach (adding fluorinated agent X to stabilize Li metal surface) is well-worn by now; the only novelty is the specific agent. In this case, it is a perfluorinated carboxylic acid. Numerous fluorinated additives for Li metal have been reported in the literature, and literally hundreds of papers peddling different additives appear every month. I feel that this article does not distinguish itself from this torrent in terms of novelty. E.g., fluorinated anhydrides have been suggested in 2020 by Lucht in JES 167 110506; even that was not too original.

The improvements in the cycling performance are on the par with other "wondrous" additives reported over the years. This claim looks comparable to other such claims, many of which turned out to be irreproducible. The performance of full cells is not stellar anyway, while the reduced capacity fade for Li/Li symmetric cells is of little import practically (as there is a lot of lithium to chew on in such cells). The authors suggest a computational rationale for their additives that it reduces the activation energy for Li deposition by (miserly) 6 kJ/mol. That's only four times the thermal energy. I do not see what this explains as the lithium is deposited anyways. It is not clear what this energy has to do with the dendrite growth. The same goes for a "DFT calculation" added to provide atomistic insight. It does not. The energy for an idealized Li atom addition to idealized surface has been computed. It is not clear what this computation has to do with anything discussed in this ms.

To me this looks like a derivative paper with no original thought or new insight, with the usual claim of superior performance that is identical to many other such claims, also without originality and insight. I do not see reasons to publish this ms in Nat Commun that stresses originality, novelty, and depth.

Other journals would be more suitable for this ms.

Reviewer #2 (Remarks to the Author):

The authors developed a surface modification strategy of Li metal anode by in-situ spontaneous reaction between Li and the organic acid. The uniform Li deposition and outstanding coulombic efficiency can be achieved by this simple treatment in convention carbonate electrolytes. The surface modification strategy enables 50-um-thin Li|high-loading-NMC811 full batteries to achieve over 300 cycles with 83.2% capacity retention under realistic testing condition. This modification strategy is very novelty and the results are very impressive. Since carbonate electrolytes is the most commonly used electrolyte in LIBs, this novel surface modification strategy is highly desirable for practical applications. The surface modification technique proposed in this work is of interest to the electrochemical community. Therefore, this manuscript is recommended for publication in Nature Communications after minor revisions.

Here are the comments in detail:

1. In the in-situ electrochemical optical microscopy tests, how to identify the magnification of optical microscopy. I have not seen the information about magnification used. So how to determine the observation magnification and calibration scale.
2. The experience detail of contact angle tests was not given in supplementary information.
3. The basic electrolyte used for evaluating the effect of the carbon chain length of the fluorinated

carboxylic acids on the stability of the Li anode should be labeled in the Figure and Figure legends in Fig.S1

4. The font of the label showing the current density (1.0 mA cm^{-2} , 0.5 mAh cm^{-2}) in Figure S2 is inconsistent with the other figures.

5. the formation cycle of Li|NMC811 full batteries should label in the figure.

Reviewer #3 (Remarks to the Author):

This work reports on the use of different fluorinated carboxylic acids to pre-treat Li metal foils for use as anode in Li batteries. Such acids are expected to have the double effect of cleaning the Li foil and creating a suitable interface. As stated in the introduction "most of the effective strategies employed to enhance the interfacial stability of Li metal anodes focused on electrolyte design". However, pre-treatment of the anode prior to battery assembly is also a common strategy and might be mentioned. The use of an acidic solution having the dual purpose of etching the native passivation layer present on the Li foil and creating an interface promoting homogeneous plating and stripping of Li is not completely new either, since a similar strategy was employed at least by Jiang et al. with iodic acid (ACS Appl. Mater. Interfaces 2017, 9, 8, 7068–7074).

Regarding the results and the discussion, the data analysis suffers several flaws.

For instance, in figure 3c reports voltage profiles during charge and discharge of NMC vs bare Li, and an arrow showing increasing charge polarization is drawn. However, a simple look at the discharge curves indicates that the problem is mostly a loss in Li inventory, which has the effect to translate the charge curves to the left in the chosen representation, rather than increasing polarization. On the other hand, in figure 3b with HFA-Li a significant increase in polarisation can be seen at discharge between cycles 3 and 50, which is not discussed at all.

The impedance spectra are reported in terms of Ohm.cm^{-2} , which is not correct since the impedance is inversely proportional to the surface area and the unit should be Ohm.cm^2 . All impedance plots have an equivalent circuit drawn in insets but the fitting of the data is not displayed. The equivalent circuit is always the same though different cell configurations were employed; for NMC vs Li it is unlikely that a good fit can be obtained with the proposed circuit. Nevertheless, for NMC-Li only the series resistance R_s is discussed, which does not require modelling with an equivalent circuit at all. It is proposed that R_s "correspond[s] to interface impedance causing (sic) by side reaction and dendrites growth" on the basis of solely ref. [20]. Attributing the variations of R_s partially to dendrite growth is controversial, since this resistance usually reflects the conductivity of the electrolyte. Likely it means that the conductivity decreases gradually due to the dissolution of side-reaction products - which indeed occurs to a larger extent with bare Li - but can hardly be linked to the growth of dendrites.

The values of the exchange current density might also be overestimated. Indeed in figure 4c the lines are taken relatively far from $\eta=0$; possibly the suitable region would be within $\pm \sim 75 \text{ mV}$, which is difficult to estimate given the resolution of the figure.

Finally, the combined effect of HFA pre-treatment and FEC and their relative efficiency is not in favor of the pre-treatment. It is mentioned that in the literature "when FEC was introduced into the electrolytes, despite the subsequent formation of a LiF-rich SEI layer, the limitations imposed by the inner SEI layer compromise the stability of Li metal anode", however in the present case FEC only is slightly more efficient (94.7%) than HFA pre-treatment only (93.71%). Though the conclusion states that it is demonstrated that "heptafluorobutyrate can simultaneously remove the surface passivation layer and construct a lithium fluoride carboxylate protective interface on Li surface.", the results show that the protection remains more imperfect than that obtained with FEC. As for the removal of the passivation layer, it does not seem clearly demonstrated by the reported results.

We thank all the reviewers for their valuable comments on our manuscript. Their constructive suggestions for improvement have certainly raised the quality of our manuscript. In order to address the reviewers' concerns, we have addressed their comments point by point.

RESPONSE TO REVIEWERS' COMMENTS

REVIEWER 1:

This is an article on surface modification of lithium metal (aka artificial solid electrolyte interphase).

Answering this journal's queries:

**There are no noteworthy results in this ms.*

**The ms has little significance due to lack of novelty.*

**The work does not provide novel, credible, or significant scientific insight.*

** The flaws of data analysis etc. are not of concern given the mundaneness of research.*

The approach (adding fluorinated agent X to stabilize Li metal surface) is well-worn by now; the only novelty is the specific agent. In this case, it is a perfluorinated carboxylic acid. Numerous fluorinated additives for Li metal have been reported in the literature, and literally hundreds of papers peddling different additives appear every month. I feel that this article does not distinguish itself from this torrent in terms of novelty. E.g., fluorinated anhydrides have been suggested in 2020 by Lucht in JES 167 110506; even that was not too original.

The improvements in the cycling performance are on the par with other "wondrous" additives reported over the years. This claim looks comparable to other such claims, many of which turned out to be irreproducible. The performance of full cells is not stellar anyway, while the reduced capacity fade for Li/Li symmetric cells is of little import practically (as there is a lot of lithium to chew on in such cells). The authors suggest a computational rationale for their additives that reduces the activation energy for Li deposition by (miserly) 6 kJ/mol. That's only four times the thermal energy. I do not see what this explains as the lithium is deposited anyways. It is not clear what this energy has to do with the dendrite growth. The same goes for a "DFT calculation" added to provide atomistic insight. It does not. The energy for an idealized Li atom addition to idealized surface has been computed. It is not clear what this computation has to do with anything discussed in this ms.

To me this looks like a derivative paper with no original thought or new insight, with the usual claim of superior performance that is identical to many other such claims, also without originality and insight. I do not see reasons to publish this ms in Nat Commun that stresses originality, novelty, and depth.

Other journals would be more suitable for this ms.

Response: We thank the reviewer for taking the time to review our manuscript. After reviewing your comments, we believe there may have been a misunderstanding of the scope and contribution of our study. We would like to take this opportunity to clarify any misunderstandings and provide additional context that may help you reconsider our submission. And we hope that this additional information will help to clarify the unique contribution of our research and the significance of HFA as an in-situ pretreatment agent for lithium metal anodes.

***Comment 1:** The approach (adding fluorinated agent X to stabilize Li metal surface) is well-worn by now; the only novelty is the specific agent. In this case, it is a perfluorinated carboxylic acid. Numerous fluorinated additives for Li metal have been reported in the literature, and literally hundreds of papers peddling different additives appear every month. I feel that this article does not distinguish itself from this torrent in terms of novelty. E.g., fluorinated anhydrides have been suggested in 2020 by Lucht in JES 167 110506; even that was not too original. The improvements in the cycling performance are on the par with other “wondrous” additives reported over the years. This claim looks comparable to other such claims, many of which turned out to be irreproducible. The performance of full cells is not stellar anyway, while the reduced capacity fade for Li/Li symmetric cells is of little import practically (as there is a lot of lithium to chew on in such cells).*

Response to comment 1: We thank the reviewer for raising these concerns and giving us the opportunity to explain the significance of our work. **To begin with, it is important to emphasize that HFA is an in-situ pretreatment agent for lithium metal anodes and not an electrolyte additive.** Electrolyte additives and lithium anode pretreatment agents have different mechanisms of action and their own advantages and disadvantages.

1. In-situ spontaneous reaction between Li and the organic acid shows a new route for practical applications of lithium metal batteries (LMBs).

Fluorinated electrolytes, additive, electrodes, and binder materials have been playing, and still have, an important role in the development of battery and especially lithium battery technologies because of their positive effects on energy and power density, life span, and safety (Fluorine and Lithium: Ideal Partners for High-Performance Rechargeable Battery Electrolytes. *Angew Chem Int Ed Engl* 2019, 58 (45), 15978-16000.)^{1,2}. The best electrolytes identified so far are based on fluorinated ether molecules rather than esters (*Science* 2022, 378 (6624))^{3,4}. The introduction of fluorine-containing

components into conventional organic carbonate electrolytes can also significantly improve the stability of lithium metal anodes (Nat Nanotechnol 2018, 13 (8), 715-722.)^{5,6}.

It would seem that significant advances in cell chemistry can be easily achieved by introducing fluorine arbitrarily into the salt, solvent/cosolvent, or functional additive structure. However, new work related to fluorinated additives has rarely been reported.

Figure 1. Literature statistical data on Li metal research in the last three years (data sources: web of science. 14th December 2022). a) Number of published works related to Li metal battery and additive. The search filter was set to retrieve papers with the following words in the topic: 'Li metal', 'battery' and 'additive'; b) Number and percentage of publications containing new fluorinated agent. The search keywords are set to "lithium metal", "battery", "additives" and " fluorinated ".

The publications on Li metal batteries were searched and summarized in Figure 1. Additive-related work accounts for a certain portion of the Li metal research work. When searching with the keywords "lithium metal", "battery", "additives" and " fluorinated ", the number of published works that could be searched in the last three years was 46. Of these published papers, 42 were related to mixed components of known fluorinated agents or mechanistic studies. Only 4 papers were related to the use of new fluorinated agents. And of these, even fewer can show significant improvement. The results of the above analysis show that although fluorination is a known route for practical applications of lithium metal batteries (LMBs), the arbitrary introduction of fluorine-containing molecules into the batteries does not easily improve the performance of the batteries and is even harmful. Therefore, we believe that the study of the in situ spontaneous reaction between Li and the organic acid shows a new way for practical applications of LMBs and will be of great interest to our colleagues.

2. Our approach is very different from the electrolyte additive strategy proposed by the reviewer,

which introduces anhydride additives into the electrolyte.

The reviewer proposes work that is considered similar to ours, using an acid anhydride species as an electrolyte additive for application in Li metal anodes. However, we still believe that the difference between the two is significant. Firstly, acid anhydride and lithium salts are two species with very different properties. Just because they both happen to have fluorine doesn't mean they are the same.

As for the electrolyte additive strategy mentioned in the review, it is the simplest way to suppress Li dendrite growth and improve the cycling stability of Li anodes. Electrolyte additives have significant practical advantages, including cost effectiveness and compatibility with the current battery industry, as no additional equipment or processes are required. However, as the electrolyte is the only component in a battery that is in contact with every other component, this means that the introduction of an additive must consider the balance of overall battery properties, ranging from bulk (e.g. ion solvation and transport and extended liquid structure) to interfacial structure and stability (e.g. preferential assembly and orientation of ions and molecules at Helmholtz planes and corrosion suppression) to interphase chemistry and morphology. It is also worth mentioning that **many of the additive-related papers do not mention the actual amount of electrolyte used in the battery, as mentioned by the reviewer of the fluorinated anhydride paper (J. Electrochem. Soc. 2020, 167 (11))**. We know that Li anodes would consume components of the electrolyte in the process of forming a stable SEI. **The excess electrolyte used in the experiment would eliminate the negative effects of electrolyte consumption and could lead to an improved effect preference.** This is one reason why the claimed effect may not be reproducible. Multi-component electrolytes also emphasise the synergistic basis of this mixture. Even small changes in these components have a significant effect on coulombic efficiency, and changes in this effect are difficult to quantify using traditional bottom-up methods. Another point to note is that the effect of a specific additive usually only works in certain electrolytes (for example, FEC does not work well in ether-based electrolytes). In general, the additive strategy for improving Li anodes has its own limitations.

The native passivation layers on the Li surface are inhomogeneous, heterogeneous in ion conduction and fail to protect the Li surface. And these passivation layers are the "seeds" that induce lithium

dendrite growth in the first place. Lithium dendrite growth has significant "self-amplifying" behaviour. Although additives can improve the stability of Li metal anodes during cycling, once the "seeds" are present, lithium dendrite formation is inevitable. As for our proposed in-situ chemical reaction method, it simultaneously removes the surface passivation layer and constructs a lithium fluoride carboxylate protective interface on the Li surface prior to cell assembly, which is a completely different strategy from adding additives to the electrolyte. And because this novel surface modification technique modifies the lithium anodes before they are assembled into batteries, it avoids the limitations of the additive strategy mentioned above.

Overall, both the species used and the strategy are completely different compared to the fluorinated anhydride additive strategy.

3. The performance of full cells is also outstanding compared to previously reported works.

The reviewer considered that the full cell performance in this work is not outstanding, and we believe that this may have been misunderstood. In order to demonstrate the significant improvement of our proposed artificial SEI strategy on the LMBs, we further compare our work with the previous works on full cell performance. As shown in the new **Table S2**, the full cell performance of our work shows longer cycle stability under more stringent conditions, indicating that our work is outstanding compared to previously reported work. **We have added this new data as Table S2 in the Supporting Information and corresponding discussions on Page 13 line 12-14 in the revised main text.**

Thus, the novel route proposed in this work proves to be useful in the design and development of stable Li metal anodes of next-generation LMBs.

Table S2. Comparison of full-cell cycling performance using different artificial SEIs.

Artificial SEI	Electrolyte	Battery Condition	Cycling Life
Adaptive "solid-liquid" interfacial protective layer (Silly Putty) ⁷	1 M LiTFSI in DOL/DME + 1wt% LiNO ₃	5 mAh cm ⁻² deposited Li 1.28 mAh cm ⁻² LFP	50 cycles
Phosphate-functionalized	0.6 M LiTFSI, 0.4 M LiBOB, 0.4 M LiF,	50 μm Li 4 mAh cm ⁻² NMC811	>300 cycles

reduced graphene oxides ⁸	0.1 M LiNO ₃ , 0.03 M LiBF ₄ , and 0.05 M LiPF ₆ in EC/DMC + 1 wt% FEC + 2 wt% VC + 3 wt% TFEC (E-3)		
PDMS coating ⁹	1 M LiPF ₆ in EC/DEC + 2 wt% VC	1 mAh cm ⁻² deposited Li LFP	100 cycles
Cu ₃ N/SBR coating ¹⁰	1 M LiPF ₆ in EC/DEC + 10 wt% FEC	10 mAh cm ⁻² deposited Li 3 mAh cm ⁻² LTO	100 cycles
Li ₃ PO ₄ SEI layer ¹¹	1 M LiPF ₆ in EC/DMC/DEC	Thick Li 0.53 mAh cm ⁻² LFP	200 cycles
Li ₂ S coating ¹²	1 M LiPF ₆ in EC/DEC	10 mAh cm ⁻² deposited Li 2.5 mAh cm ⁻² LFP	150 cycles
Reactive polymer composite (RPC) ¹³	1 M LiPF ₆ in EC/EMC + 2 wt% LiBOB	1.9-fold excess Li in a 3D host 3.4 mAh cm ⁻² NMC532	200 cycles
Cation-Tethered Flowable Polymer ¹⁴	2 M LiTFSI + 2 M LiDFOB in DME	25 μm Li 2.7 mAh cm ⁻² NMC532	70 cycles
LiAl-FBD coating ¹⁵	1 M LiPF ₆ in EC/DEC + 10% FEC	50 μm Li 2.6 mAh cm ⁻² NMC811	180 cycles
This work: In-situ spontaneous reaction coating HFA-Li	1 M LiPF₆ in EC/EMC+ 5 wt% FEC	50 μm Li 4.0 mAh cm⁻² NMC811	300 cycles

Page 13 line 12-14 in revised main text:

The observed rapid capacity decay is indicative of the instability of the interface between the Bare-Li anode and electrolyte. **In particular, the performance of the HFA-Li||NMC811 full cell fabricated in this study is better than that of previously reported Li||NMC811 full cells in carbonate- and ether-based electrolytes (Table S2).**

EIS measurements were also performed to evaluate the change in the electrode/electrolyte interface properties of Bare-Li and HFA-Li after a long-term cycle.

Comment 2: *The authors suggest a computational rationale for their additives that it reduces the activation energy for Li deposition by (miserly) 6 kJ/mol. That's only four times the thermal energy.*

I do not see what this explains as the lithium is deposited anyways. It is not clear what this energy has to do with the dendrite growth. The same goes for a “DFT calculation” added to provide atomistic insight. It does not. The energy for an idealized Li atom addition to idealized surface has been computed. It is not clear what this computation has to do with anything discussed in this ms.

Response to comment 2: We thank the reviewer for raising these concerns and giving us the opportunity to explain the measurement of the activation energy for Li deposition and the DFT calculation. We have also added additional theoretical calculation and related description to provide further atomistic insight.

To measure the activation energy for Li deposition, the use of temperature-dependent electrochemical impedance spectroscopy (EIS) to determine the kinetics of different interfacial processes was first proposed by Prof. Kang Xu.¹⁶ The activation energy represent the energy barrier for Li deposition, which is contributed by several processes including the breaking-up of the Li⁺ solvent sheath, the diffusion of Li⁺ through SEI and the redox process of Li⁺. The value of this activation energy is typically in the order of tens of kJ per mole.^{16, 17, 18} And a change of a few to a dozen kilojoules per mole of activation energy after HFA treatment is reasonable according to previous reports, because there is no change in the solvent structure (related to the composition of the electrolyte) as well as in the redox process of lithium.^{16, 17, 18} Lithium is deposited anyway, as the reviewer said, and if lithium cannot be deposited, then the point of the study is lost. Our strategy is to regulate the Li deposition behavior. And the reduction of the energy barrier for Li deposition indicates the lithiophilic character of the HFA-treated Li anode. It indicates that Li deposition becomes easier on the HFA-treated Li anode and the HFA-Li interface can effectively regulate the Li deposition behaviour as visualised by the SEM and in situ electrochemical optical microscopy tests.

As for DFT calculations, they are limited by current computing power. **In fact, all current theoretical calculations are modelled based on ideal surfaces and the number of atoms calculated in the model is also limited**, which is the prevailing bottleneck of theoretical calculations. For example, **the ideal lithium surface is commonly used for interfacial mechanics studies** (J. Power Sources 2019, 438. Adv. Energy Mater. 2019, 9 (42). Joule 2019, 3 (3), 732-744.)^{19, 20, 21}. However, it cannot be denied that DFT computations can provide theoretical insights

for various battery material systems.

Also, previous work had reported that the strong adsorption energy of the protective layer material with Li surface can limit the longitudinal growth of Li and facilitate the lateral growth of Li to form a flat pie-type morphology, thus avoiding the formation of Li dendrites.²²

In our works, it contributed significantly to the understanding of the intrinsic mechanism of the HFA-Li interface for improving the stability of Li anode.

The PDOS (Projected density of state) of adsorbed Li on Li substrate with HFA-Li and BA-Li were also evaluated as shown in new Figure 4c, d. when HFA-Li was introduced, the O atom of HFA-Li interacted significantly with the Li adatom compared to the introduction of BA-Li. The high orbital hybridization of Li-s and O-2p states indicates the strong interactions between HFA-Li and adsorbed Li on Li substrate. **The new data and description have been added in the revised paper (Page 15 line 8-17) and the related changes are marked in yellow color.**

Figure 4. Stable configurations and corresponding charge density differences of (a) BA-Li and (b) HFA-Li on Li (100) surface. The brown, red, pink, green, gray, and purple balls represent C, O, H, Li, F, and adsorbed Li atoms, respectively. The yellow and blue regions represent charge accumulation and loss, respectively; The corresponding PDOS of the adsorbed Li atoms and its nearest neighbouring O atom of HFA-Li (c) and Bare-Li (d). (e) Contact angle of Bare-Li and HFA-Li; (f) Optical images of the in-situ Li deposition on Bare-Li and HFA-Li at a deposition current of

5 mA cm⁻²; (g) Tafel plot of HFA-Li and Bare-Li; EIS plots of the Li/Li symmetric cells containing (h) Bare-Li and (i) HFA-Li at different temperatures before cycling; (j) Activation energy (E_a) of HFA-Li and Bare-Li. The inset shows the Arrhenius behavior of the resistant.

In general, theoretical calculations can be used to provide possible intrinsic mechanisms at the atomic level. The theoretical calculations similar to ours have also been reported in recent years. In the DFT calculation of our work, strong interactions between HFA-Li and Li substrate and the change of charge distribution were proved, which can be well support the improved lithiophilic character of the HFA-Li interface and ensuring uniform Li⁺ ion flux to suppress Li dendritic growth during cycling

Page 15 line 8-17 in revised main text:

To probe the critical effect of HFA-Li on uniform lithium-ion flux and the lithiophilic performance of Li metal anode, the interactions of HFA-Li, BA-Li, and single Li ion with the Li substrate were evaluated through density functional theory (DFT) calculations. **Figure 4a, b and Figure S10** shows the stable configurations and corresponding charge density difference of HFA-Li, BA-Li and Li ion on the surface of Li (100). The blue and yellow regions represent charge loss and accumulation, respectively (**Figure 4a, b**). The adsorption of HFA-Li has a significant impact on the electronic state and charge distribution of the surrounding area, leading to an electron transfer from Li to O atoms surrounding the region.²³ The adsorption energies of HFA-Li and BA-Li on a Li substrate were determined to be -2.11 eV and -1.99 eV, respectively, after fully optimizing the structure. For comparison, the Li adsorption energy of the Li substrate was lower at -1.61 eV (**Figure S10**). The PDOS (Projected density of state) of adsorbed Li on Li substrate with HFA-Li and BA-Li were also evaluated based on DFT calculations (**Figure 4c, d**). When HFA-Li was introduced, the O atom of HFA-Li interacted significantly with the Li adatom compared to the introduction of BA-Li. The high orbital hybridization of Li-s and O-2p states indicates the strong interactions between HFA-Li and adsorbed Li on Li substrate. The adsorption energy of the Li substrate for HFA-Li was also higher than that for BA-Li. This can be attributed to the strong electron-withdrawing effect of the C-F functional groups, which further promotes the electron

transfer and enhances the HFA-Li adsorption capacity.²⁴ And the strong adsorption energy of HFA-Li with Li surface can suppress the longitudinal growth of Li and facilitate the lateral growth of Li to achieve uniform Li deposition.²² Owing to their strong interaction with the Li substrate, the O atoms in the carboxylic acid group can act as nucleation sites for Li deposition to homogenize Li ion flux, promote uniform Li deposition, and consequently improve the lithiophilic character of the interface.^{23, 25} Therefore, the carboxyl and C-F functional groups of HFA-Li were possibly critical to ensure uniform Li⁺ ion flux and Li deposition, and suppress Li dendritic growth during cycling.

REVIEWER 2:

Reviewer #2 (Remarks to the Author):

The authors developed a surface modification strategy of Li metal anode by in-situ spontaneous reaction between Li and the organic acid. The uniform Li deposition and outstanding coulombic efficiency can be achieved by this simple treatment in convention carbonate electrolytes. The surface modification strategy enables 50-um-thin Li|high-loading-NMC811 full batteries to achieve over 300 cycles with 83.2% capacity retention under realistic testing condition. This modification strategy is very novelty and the results are very impressive. Since carbonate electrolytes is the most commonly used electrolyte in LIBs, this novel surface modification strategy is highly desirable for practical applications. The surface modification technique proposed in this work is of interest to the electrochemical community. Therefore, this manuscript is recommended for publication in Nature Communications after minor revisions.

Here are the comments in detail:

1. In the in-situ electrochemical optical microscopy tests, how to identify the magnification of optical microscopy. I have not seen the information about magnification used. So how to determine the observation magnification and calibration scale.
2. The experience detail of contact angle tests was not given in supplementary information.
3. The basic electrolyte used for evaluating the effect of the carbon chain length of the fluorinated carboxylic acids on the stability of the Li anode should be labeled in the Figure and Figure legends in Fig.S1
4. The font of the label showing the current density (1.0 mA cm⁻², 0.5 mAh cm⁻²) in Figure S2 is inconsistent with the other figures.
5. the formation cycle of Li|NMC811 full batteries should label in the figure.

Response: Thank you very much for your positive comments!

Comment 1: In the in-situ electrochemical optical microscopy tests, how to identify the magnification of optical microscopy. I have not seen the information about magnification used. So how to determine the observation magnification and calibration scale.

Response to comment 1: Thank you for your important and valuable suggestions. The magnification of the optical microscope can be calibrated by the thickness of the lithium foil. This ensures that the magnification is the same for each experiment. In the in-situ electrochemical optical microscopy experiment, the thickness of the lithium foil used was 0.5 mm, which was added to the experimental details.

Comment 2: The experience detail of contact angle tests was not given in supplementary information.

Response to comment 2: Thank you for your careful check. The details of the contact angle test have been given in the supplementary information.

Comment 3: The basic electrolyte used for evaluating the effect of the carbon chain length of the fluorinated carboxylic acids on the stability of the Li anode should be labeled in the Figure and Figure legends in Fig.S1

Response to comment 3: Thank you for your attention to detail. The basic electrolytes used are labelled in the figure and figure legends in Fig.S1.

Figure S1. Voltage–time curves of the Li/Li symmetric cells composed of Li anodes treated with fluorinated carboxylic acids of different carbon chain lengths in a 1 M LiPF₆ in EC/EMC (v/v = 3:7) electrolyte at a current density of 1.0 mA cm⁻² and a capacity of 0.5 mAh cm⁻². DFA-Li, PFA-Li, HFA-Li, PFVA-Li, and UFA-Li correspond to Li anode treated with difluoroacetic acid, pentafluoropropionic acid, heptafluorobutyric acid, perfluorovaleric acid, and undecafluorohexanoic acid, respectively.

Comment 4: The font of the label showing the current density (1.0 mA cm⁻², 0.5 mAh cm⁻²) in Figure S2 is inconsistent with the other figures.

Response to comment 4: Thank you for your careful check. The font of the label has been corrected.

Figure S2. Voltage–time curves of the Li/Li symmetric composed of BA-Li and HFA-Li electrodes in a 1 M LiPF₆ in EC/EMC (v/v = 3:7) electrolyte at a current density of 1.0 mA cm⁻² and a capacity of 0.5 mAh cm⁻².

Comment 5: The formation cycle of Li|NMC811 full batteries should label in the figure.

Response to comment 5: The formation cycle of Li|NMC811 full batteries has been labeled in the Figure 3a.

Figure 3. (a) Long-cycling performance of the Li||NMC811 cell.

REVIEWER 3:

This work reports on the use of different fluorinated carboxylic acids to pre-treat Li metal foils for use as anode in Li batteries. Such acids are expected to have the double effect of cleaning the Li foil and creating a suitable interface. As stated in the introduction "most of the effective strategies employed to enhance the interfacial stability of Li metal anodes focused on electrolyte design". However, pre-treatment of the anode prior to battery assembly is also a common strategy and might be mentioned. The use of an acidic solution having the dual purpose of etching the native passivation layer present on the Li foil and creating an interface promoting homogeneous plating and stripping of Li is not completely new either, since a similar strategy was employed at least by Jiang et al. with iodic acid (ACS Appl. Mater. Interfaces 2017, 9, 8, 7068–7074).

Regarding the results and the discussion, the data analysis suffers several flaws.

For instance, in figure 3c reports voltage profiles during charge and discharge of NMC vs bare Li, and an arrow showing increasing charge polarization is drawn. However, a simple look at the discharge curves indicates that the problem is mostly a loss in Li inventory, which has the effect to translate the charge curves to the left in the chosen representation, rather than increasing polarization. On the other hand, in figure 3b with HFA-Li a significant increase in polarisation can be seen at discharge between cycles 3 and 50, which is not discussed at all.

The impedance spectra are reported in terms of Ohm.cm^{-2} , which is not correct since the impedance is inversely proportional to the surface area and the unit should be Ohm.cm^2 . All impedance plots have an equivalent circuit drawn in insets but the fitting of the data is not displayed. The equivalent circuit is always the same though different cell configurations were employed; for NMC vs Li it is unlikely that a good fit can be obtained with the proposed circuit. Nevertheless, for NMC-Li only the series resistance R_s is discussed, which does not require modelling with an equivalent circuit at all. It is proposed that R_s "correspond[s] to interface impedance causing (sic) by side reaction and dendrites growth" on the basis of solely ref. [20]. Attributing the variations of R_s partially to dendrite growth is controversial, since this resistance usually reflects the conductivity of the electrolyte. Likely it means that the conductivity decreases gradually due to the dissolution of side-reaction products - which indeed occurs to a larger extent with bare Li - but can hardly be linked to the growth of dendrites.

The values of the exchange current density might also be overestimated. Indeed in figure 4c the lines are taken relatively far from $\eta=0$; possibly the suitable region would be within $\pm \sim 75\text{mV}$, which is difficult to estimate given the resolution of the figure.

Finally, the combined effect of HFA pre-treatment and FEC and their relative efficiency is not in favor of the pre-treatment. It is mentioned that in the literature "when FEC was introduced into the electrolytes, despite the subsequent formation of a LiF-rich SEI layer, the limitations imposed by the inner SEI layer compromise the stability of Li metal anode", however in the present case FEC only is slightly more efficient (94.7%) than HFA pre-treatment only (93.71%). Though the conclusion states that it is demonstrated that "heptafluorobutyrate can simultaneously remove the surface passivation layer and construct a lithium fluoride carboxylate protective interface on Li surface.", the results show that the protection remains more imperfect than that obtained with FEC. As for the removal of the passivation layer, it does not seem clearly demonstrated by the reported results.

Response: Thank you very much for your many important and valuable suggestions, and we have added extensive new experiments and theoretical simulations to fully confirm our hypothesis. We

hope the revised manuscript is now suitable for publication in Nature Communications.

Comment 1: *This work reports on the use of different fluorinated carboxylic acids to pre-treat Li metal foils for use as anode in Li batteries. Such acids are expected to have the double effect of cleaning the Li foil and creating a suitable interface. As stated in the introduction "most of the effective strategies employed to enhance the interfacial stability of Li metal anodes focused on electrolyte design". However, pre-treatment of the anode prior to battery assembly is also a common strategy and might be mentioned. The use of an acidic solution having the dual purpose of etching the native passivation layer present on the Li foil and creating an interface promoting homogeneous plating and stripping of Li is not completely new either, since a similar strategy was employed at least by Jiang et al. with iodic acid (ACS Appl. Mater. Interfaces 2017, 9, 8, 7068–7074).*

Response to comment 1: Thank you very much for your important and valuable suggestions. We have carefully read the references mentioned in the review comment (ACS Appl. Mater. Interfaces 2017, 9, 8, 7068–7074.), **and the additional description of the iodic acid pretreatment strategy has been added in the introduction part of the manuscript.** We hope that these newly added descriptions can help our colleagues to understand our progress in the field of anode pretreatment. In brief, our proposed pretreatment strategy overcomes some of the intractable drawbacks of the iodic acid strategy and enables remarkable cycle stability of LMBs.

As we known, the passivation layer on the Li surface is detrimental to the cycling stability of the Li anode and cannot be removed from the Li surface even with the most efficient electrolyte design. Iodic acid has been proposed and successfully removes the passivation layer by spontaneous reaction (ACS Appl. Mater. Interfaces 2017, 9, 8, 7068–7074.). **However, one of the major problems of this strategy is the shuttle effect of iodide ions, which hinders its extensive application in LMBs.** First, the reaction products I^- and IO_3^- are highly soluble in the electrolyte until saturation.²⁶ During the charging process, I^- can be oxidized to I_3^- at the cathode side (3.55V, vs. Li^+/Li) and I_3^- is also soluble in the electrolyte.^{26, 27} The oxidized product, I_3^- , would migrate to the Li anode side driven by the concentration gradient or electric field and react spontaneously with lithium. This means that the iodic acid pre-treatment strategy is not compatible with high voltage cathode materials such as Ni-rich $LiNi_xMn_yCo_{1-x-y}O_2$ layered cathode materials (the charge cut-off voltage reaches 4.30V). The irreversible self-discharge behavior during the charging process will occur and continuously consume the Li reservoir if paired with high voltage cathode materials. As a result, the authors eventually chose sulfur cathode (charge cut-off voltage is lower than 3.0V) and

thick Li foil (excess Li sources) for the full cell performance tests.

Reaction of Iodic acid pre-treatment:

The Cathode side:

The Anode side:

In general, iodic pre-treatment can successfully remove the passivation layer of the Li anode, but cannot be used in high voltage battery systems due to the shuttle effect of I_3^-/I^- . It is also important to note that the halogen content in commercial batteries is also limited to ppm levels to prevent degradation of battery lifespan. Compared to the iodic acid pre-treatment strategy, our proposed strategy successfully removes the passivation layer while overcoming the drawbacks caused by the introduction of halogen.

Page 3 line 1-5 in revised main text:

During spontaneous chemical reactions, the chemical composition of the passivation layer along the Li surface varies. Consequently, the electrochemical kinetics also changes at different positions along the metal anode, yielding a non-uniform Li-ion flux and triggering the growth of Li dendrites.⁸ Iodic acid has been proposed and successfully removes the passivation layer by spontaneous reaction.²⁸ However, the shuttle effect of iodide ions hinders its extensive application in LMBs. The irreversible self-discharge behavior during the charging process will occur and continuously consume the Li reservoir if paired with high voltage cathode materials.^{26, 27}

In this study, a novel strategy to rebuild the interfacial layer on the Li surface and simultaneously construct a protective layer with lithiophilic properties is proposed (**Scheme 1**).

And importantly, this HFA pre-treatment was successfully applied to 50- μm -thin Li||high-loading-NMC811 full batteries to achieve an outstanding cycle stability. To further demonstrate the

advantage of our proposed strategy, the comparison of full-cell cycling performance using different artificial SEIs was added in supporting information as mentioned in the response to reviewer 1. Li||NMC811 full cells assembled using the HFA-Li anode exhibit considerably improved cycle stability compared to previous reported works as shown in **Table S2**.

Table S2. Comparison of full-cell cycling performance using different artificial SEIs.

Artificial SEI	Electrolyte	Battery Condition	Cycling Life
Adaptive “solid-liquid” interfacial protective layer (Silly Putty) ⁷	1 M LiTFSI in DOL/DME + 1wt% LiNO ₃	5 mAh cm ⁻² deposited Li 1.28 mAh cm ⁻² LFP	50 cycles
Phosphate-functionalized reduced graphene oxides ⁸	0.6 M LiTFSI, 0.4 M LiBOB, 0.4 M LiF, 0.1 M LiNO ₃ , 0.03 M LiBF ₄ , and 0.05 M LiPF ₆ in EC/DMC + 1 wt% FEC + 2 wt% VC + 3 wt% TFEC (E-3)	50 μm Li 4 mAh cm ⁻² NMC811	>300 cycles
PDMS coating ⁹	1 M LiPF ₆ in EC/DEC + 2 wt% VC	1 mAh cm ⁻² deposited Li LFP	100 cycles
Cu ₃ N/SBR coating ¹⁰	1 M LiPF ₆ in EC/DEC + 10 wt% FEC	10 mAh cm ⁻² deposited Li 3 mAh cm ⁻² LTO	100 cycles
Li ₃ PO ₄ SEI layer ¹¹	1 M LiPF ₆ in EC/DMC/DEC	Thick Li 0.53 mAh cm ⁻² LFP	200 cycles
Li ₂ S coating ¹²	1 M LiPF ₆ in EC/DEC	10 mAh cm ⁻² deposited Li 2.5 mAh cm ⁻² LFP	150 cycles
Reactive polymer composite (RPC) ¹³	1 M LiPF ₆ in EC/EMC + 2 wt% LiBOB	1.9-fold excess Li in a 3D host 3.4 mAh cm ⁻² NMC532	200 cycles
Cation-Tethered Flowable Polymer ¹⁴	2 M LiTFSI + 2 M LiDFOB in DME	25 μm Li 2.7 mAh cm ⁻² NMC532	70 cycles
LiAl-FBD coating ¹⁵	1 M LiPF ₆ in EC/DEC + 10% FEC	50 μm Li 2.6 mAh cm ⁻² NMC811	180 cycles
This work: In-situ spontaneous reaction coating HFA-Li	1 M LiPF₆ in EC/EMC + 5 wt% FEC	50 μm Li 4.0 mAh cm⁻² NMC811	300 cycles

Page 13 line 12-14 in revised main text:

The observed rapid capacity decay is indicative of the instability of the interface between the Bare-Li anode and electrolyte. In particular, the performance of the HFA-Li||NMC811 full cell fabricated in this study is better than that of previously reported Li||NMC811 full cells in carbonate- and ether-based electrolytes (Table S2).

EIS measurements were also performed to evaluate the change in the electrode/electrolyte interface properties of Bare-Li and HFA-Li after a long-term cycle.

Comment 2: Regarding the results and the discussion, the data analysis suffers several flaws. For instance, in figure 3c reports voltage profiles during charge and discharge of NMC vs bare Li, and an arrow showing increasing charge polarization is drawn. However, a simple look at the discharge curves indicates that the problem is mostly a loss in Li inventory, which has the effect to translate the charge curves to the left in the chosen representation, rather than increasing polarization. On the other hand, in figure 3b with HFA-Li a significant increase in polarisation can be seen at discharge between cycles 3 and 50, which is not discussed at all.

Response to comment 2: Thank you for your careful consideration. We are very sorry for the confusion caused by our arrow labelling error in Figure 3c. In the revised manuscript, the arrow labelling has been corrected to "Depletion of Li reservoir" in the new Figure 3c. The description of the increase in polarization with HFA-Li in Figure 3b has also been added to the revised main text on page 13, lines 1-7. It's indeed necessary to add this missing part of the description. The related changes are highlighted in yellow in the revised manuscript.

Figure 3. (c) Charge–discharge curves of Bare-Li taken at the 3rd, 50th, 70th, 80th, 90th and 100th cycle of the full cell testing.

Page 13 line 1-7 in revised main text:

The HFA-Li||NMC811 full cell demonstrated a stable cycle of over 300 cycles with 83.2% capacity retention. Although the electrolyte-induced resistive SEI built up during Li plating/stripping, leading to an increase in polarization, especially in the early stages of battery cycling, HFA-Li homogenized the Li ion flux, which in turn induced uniform Li deposition and minimized the occurrence of tortuous lithium dendrites. As a result, no significant capacity drop or sudden cell death was observed during cycling. In contrast, the unstable interface between Bare-Li and the electrolyte accelerated the exposure and growth of fresh lithium dendrites, leading to rapid depletion of the Li reservoir. The capacity of the Bare-Li||NMC811 full cell declined rapidly after only 50 cycles.

Comment 3: The impedance spectra are reported in terms of Ohm.cm⁻², which is not correct since the impedance is inversely proportional to the surface area and the unit should be Ohm.cm². All impedance plots have an equivalent circuit drawn in insets but the fitting of the data is not displayed. The equivalent circuit is always the same though different cell configurations were employed; for NMC vs Li it is unlikely that a good fit can be obtained with the proposed circuit. Nevertheless, for NMC-Li only the series resistance R_s is discussed, which does not require modelling with an equivalent circuit at all. It is proposed that R_s "correspond[s] to interface impedance causing (sic) by side reaction and dendrites growth" on the basis of solely ref. [20]. Attributing the variations of R_s partially to dendrite growth is controversial, since this resistance usually reflects the conductivity of the electrolyte. Likely it means that the conductivity decreases gradually due to the dissolution of side-reaction products - which indeed occurs to a larger extent with bare Li - but can hardly be linked to the growth of dendrites.

Response to comment 3: Thank you very much for your careful check and valuable suggestions. This comment is very helpful for us to improve our paper. We have carefully made the refinements as suggested by the reviewer.

The unit problem "Ohm·cm⁻²" of impedance spectra has been fixed to "Ohm·cm²". And we agreed with the reviewer that R_s does not require equivalent circuit fitting. So, the equivalent circuit model of Li||NMC811 full cells has been removed, as suggested by the reviewer.

Figure 3. Electrochemical impedance spectroscopy (EIS) tests of the Li||NMC811 cells after 2(e), 50(f), and 100(g) cycle of the full cell testing

However, we may disagree with the reviewer's view on the source of the R_s . As for R_s , we maintain that the R_s is strongly related to the growth of Li dendrites. Additional supporting reference (Nature Energy 2021, 6 (7), 723-732.) and experiment have been added. As mentioned by the reviewer, impedance is inversely proportional to surface area. The growth of Li dendrites greatly increases the specific surface area of the Li anode. And this should lead to a sharp decrease in impedance or even R_s approaching 0. However, this inference does not take into account the limited electrolyte in the cell and the swallowing of the electrolyte by a large number of lithium dendrite interstices. The trace amount of electrolyte cannot fully saturate the dramatically increased Li surface area. **As more 'pores' are generated within the cycled Li, more and more liquid electrolyte flows freely into these new areas and is separated from the bulk electrolyte.** The reduction in electrolyte contact eventually leads to a reduction in ion transport channels and is reflected in an increase in R_s .⁴ To corroborate the close correlation between R_s and Li dendrite growth, We also tested the impedance evolution of Li/Li symmetric cells over time. As we known, Li dendrites would not generate during cell resting process, and only interfacial side reactions would occur. As a result, no increase in R_s is observed for either HFA-Li or Bare-Li assembled symmetric cells. Additional supporting reference (Nature Energy 2021, 6 (7), 723-732.) have also been added to the revised paper.

Figure S8. The impedance evolution of (a) Bare-Li | Bare-Li and (b) HFA-Li | HFA-Li symmetric cells over time in a 1 M LiPF₆ in EC/EMC (v/v = 3:7) electrolyte with 5wt% FEC.

Comment 4: The values of the exchange current density might also be overestimated. Indeed in figure 4c the lines are taken relatively far from $\eta=0$; possibly the suitable region would be within ± 75 mV, which is difficult to estimate given the resolution of the figure.

Response to comment 4: Thank you so much for your careful check. We are sorry that the resolution of the figure may have caused you some confusion. We have improved the resolution and format of the Figure 4e. And the linear fitting region is taken to be within ± 75 mV as suggested by the reviewer. The value of i^0 has been recalculated and, as predicted by the reviewer, is lower than the previous value (Bare-Li from 0.341 mA cm^{-2} to 0.150 A cm^{-2} and HFA-Li from 1.178 A cm^{-2} to 0.463 A cm^{-2}). The value of i^0 has been updated in the revised paper.

Figure 4e. Tafel plot of HFA-Li and Bare-Li

Page 17 line 2-5 in revised main text:

The exchange current density is obtained from the Tafel equation.²¹ The value of i^0 of HFA-Li (0.463 mA cm⁻²) is higher than that of Bare-Li (0.150 mA cm⁻²), which implies that the charge transfer processes are faster at the electrode/electrolyte interface of HFA-Li than those on Bare-Li.

Comment 5: Finally, the combined effect of HFA pre-treatment and FEC and their relative efficiency is not in favor of the pre-treatment. It is mentioned that in the literature "when FEC was introduced into the electrolytes, despite the subsequent formation of a LiF-rich SEI layer, the limitations imposed by the inner SEI layer compromise the stability of Li metal anode", however in the present case FEC only is slightly more efficient (94.7%) than HFA pre-treatment only (93.71%). Though the conclusion states that it is demonstrated that "heptafluorobutyrate can simultaneously remove the surface passivation layer and construct a lithium fluoride carboxylate protective interface on Li surface.", the results show that the protection remains more imperfect than that obtained with FEC.

Response to comment 5: Thank you very much for your valuable suggestions. To begin with, it is important to emphasise that **HFA is an in-situ pretreatment agent for lithium metal anodes and not an electrolyte additive.** Electrolyte additives and lithium anode pretreatment agents have different mechanisms of action and their own advantages and disadvantages.

To further examine the protective feature of the HFA-Li coating, the impedance evolution of Li/Li symmetric cells over time was tested as mentioned above. In 1 M LiPF₆ in EC/EMC (v/v = 3:7) with 5.0 wt% FEC, the HFA-Li symmetric cell exhibited a low and stable interfacial impedance over the entire cell resting time. In contrast, despite the introduction of FEC into the electrolyte, the impedance of the Li/Li symmetric cell assembled using the Bare-Li electrodes increase sharply after the cell assembling to 32 h (from ~110 Ohm·cm² to ~190 Ohm·cm²). This result demonstrates the electrolyte-blocking feature of the HFA-Li coating. Also, FEC does not prevent the impedance increase and parasitic reactions occur on the surface of Bare-Li during the resting process.

Figure S8. The impedance evolution of (a) Bare-Li | Bare-Li and (b) HFA-Li | HFA-Li symmetric cells over time in a 1 M LiPF₆ in EC/EMC (v/v = 3:7) electrolyte with 5 wt% FEC.

The above result also shows that electrolyte additives (FEC in our case) and lithium anode pretreatment agents (HFA in our case) have their own advantages. The native passivation layers on the Li surface are inhomogeneous, heterogeneous in ionic conduction, and fail to protect the Li surface even with FEC. And these passivation layers have been the "seeds" that induce the growth of lithium dendrites at the outset. **Lithium dendrite growth has significant "self-amplifying" behavior. Although FEC additives can enhance the stability of Li metal anodes during cycling, once the "seeds" is there, it inevitably leads to lithium dendrite formation.** As for the lithium anode pre-treatment agents of HFA, it simultaneously removes the surface passivation layer and construct a protective interface on Li surface before cell assembly. However, despite having the protection of artificial SEI layer, it is still unavoidably for Li anode to form some protuberant tips due to various fluctuations in the system during long-term Li plating/stripping. And unlike electrolyte additives, lithium anode pretreatment agents can not repair SEI during cycling. We demonstrated the synergistic effects of HFA pre-treatment and FEC additive, which enables Li anodes to successfully achieve excellent cycling stability and coulombic efficiency. The sole application of HFA can also remarkably improve the cycling stability of Li metal anodes (the CE increase from 78.93% to 93.71%). As mentioned by the reviewer, the CE value of lithium metal pretreated with HFA was slightly lower than that of FEC additive alone (94.7%). However, compared to many other reported organic/inorganic additives, the Li metal anode pretreated with HFA has a more stable cycling performance.^{29, 30, 31, 32, 33, 34} **And we still believe that it is not appropriate to compare electrolyte additives with pre-treatment strategy. Because HFA pre-treatment strategy has its own unique mechanism that can be used well in conjunction with additive methods.**

We hope that our answers will address your concerns. In general, this simple molecule, HFA, with its unique mechanism can effectively promote homogeneous plating and stripping of Li and inhibit

the growth of lithium dendrites. We believe that the findings of this work are meaningful for our field of research.

Page 10 line 4-11 in revised main text:

Consequently, dead Li⁰ causing by isolated Li particles trapped in the SEI during Li stripping were significantly reduced.

Li/Li symmetric cells were first used to evaluate the electrolyte-blocking feature of the HFA-Li coating. **Figure S8** shows the impedance evolution of Bare-Li/Bare-Li and HFA-Li/HFA-Li symmetric cells over time in a 1 M LiPF₆ in EC/EMC (v/v = 3:7) electrolyte with 5% FEC. the HFA-Li symmetric cell exhibited a low and stable interfacial impedance over the entire cell resting time. In contrast, the impedance of the Li/Li symmetric cell assembled using the Bare-Li electrodes increase sharply after the cell assembling to 32 h (from ~110 Ohm·cm² to ~190 Ohm·cm²).

The long-term cycle stabilities of Li/Li symmetric cells assembled using the HFA-Li and Bare-Li electrodes were also evaluated.

***Comment 6:** As for the removal of the passivation layer, it does not seem clearly demonstrated by the reported results.*

Response to comment 6: Thank you very much for your important and valuable suggestions. We have added extensive experiments and theoretical calculation to demonstrate the removal of the passivation layer by HFA pre-treatment.

1. The spontaneous reaction between the main components of passivation layer (Li₂CO₃, Li₂O, LiOH) and heptafluorobutyric acid (HFA) was first investigated by theoretical calculations.

First, the free energy for the reactions between HFA and the main components of the passivation layer (Li₂CO₃, Li₂O, LiOH) were calculated by VASP5.4.1 program. VASPsol program is used to take solvation effect into consideration, and the solvent is water whose dielectric constant is 78.4.³⁵ KPOINTS is generated by VASPKIT program³⁶. The K-Mesh is Monkhorst-Pack Scheme³⁷, and the resolved value is 0.025. The single molecule is in the cube cell with a side length of 25 Å. The Gibbs free energy is obtained by vibration analysis and statistical thermodynamics, and the temperature is 298.15 K. The calculated reaction equation and the calculated ΔG , are listed below.

$$\Delta G_r = -197.068 \text{ kJ mol}^{-1}$$

$$\Delta G_r = -415.621 \text{ kJ mol}^{-1}$$

$$\Delta G_r = -278.756 \text{ kJ mol}^{-1}$$

The results of the calculated reaction free energy showed that the reaction of HFA with Li_2CO_3 , Li_2O , and $LiOH$ are thermodynamically spontaneous.

2. Direct reaction experiments demonstrated that HFA can spontaneously react with the passivation layer.

Figure. S1 XRD patterns of $LiOH$, Li_2CO_3 and Li_2O before and after reaction with excess HFA.

After the theoretical proof, experimental verification of the above reaction equation was carried out. XRD (X-ray diffraction) was employed to characterize the changes of reactants before and after the reaction. According to the reaction equations, Li_2CO_3 , Li_2O , $LiOH$ were first reacted with excess HFA, and the reacted products were dried to remove H_2O and any residual HFA. The XRD patterns of $LiOH$, Li_2CO_3 and Li_2O before and after reaction with excess HFA are shown in new Figure S1. The characteristic peaks of $LiOH$, Li_2CO_3 and Li_2O all disappeared completely after the reaction,

and the characteristic peaks of the XRD patterns of all reaction products were consistent and identical, indicating the production of HFA-Li

3. The TOF-SIMS test result directly shows that the passivation layer on Li surface was removed after HFA treatment.

Figure. S2 The TOF-SIMS depth sputter curves of Li surface before and after HFA treatment

TOF-SIMS was used to detect changes in the components of the Li surface before and after HFA treatment. The new Figure. S2a shows the components of the Li surface with sputtering time before HFA treatment. The signal of LiO_2H_2^- and LiO_2H^- can be attributed to the contribution from LiOH. And the strong signals of LiO_2H_2^- and LiO_2H^- suggest that the component of the passivation layer on Li surface is dominated by LiOH. An obvious signal of LiO^- can also be observed. The signal of LiO^- is from LiOH or Li_2O . And the signal of LiCO_3^- usually comes from Li_2CO_3 . The above results show that lithium is covered with a passivation layer composed of Li_2CO_3 , LiOH, and Li_2O , and LiOH is the main component of the passivation layer. And there is also a small amount of Li_2O that is mixed in with the lithium metal. This result is consistent with the previous work.³⁸

After HFA treatment, the signal coming from the component of LiOH has been significantly reduced (Figure. S2b). And the signal of LiO^- can be attributed to the contribution of Li_2O and lithium heptafluorobutyrate. More importantly, a series of ionic fragments appear corresponding to the formation of lithium heptafluorobutyrate. The above results directly demonstrate that HFA pre-treatment can successfully remove the surface passivation layer and construct a protective lithium fluorinatedcarboxylate interface on the Li surface.

(It should be noted that TOF-SIMS can only be used to detect changes in components and cannot be used to compare the thickness of different coatings. This is because TOF-SIMS analysis has a selected sputtering property, i.e., the sputtering rate is different for different component surfaces. Typically, TOF-SIMS has a faster etch rate for organic materials than for inorganic materials.)

We hope that the newly added experiments and theoretical calculations will address your concerns.

The corresponding description and figure mentioned above have been added and highlighted in yellow in the manuscript and in the supplementary information.

Page S2 line 5-7 in revised supplementary information:

Electrolytes preparation were performed in the glove box as well. Li_2CO_3 (99.9%) Li_2O (99.9%) and LiOH (99.9%) were acquired from Macklin and used as received without any further processing.

Page 4 line 20 to page 5 line 3 in revised main text:

thus, fluorinated carboxylic acid can react with Li_2CO_3 and even with the strong base LiOH and basic Li_2O on the Li surface according to the following reaction (in the supplementary information, an in-depth examination is provided, showing both experimental and theoretical calculations to illustrate the removal of the passivation layer.(Figure S1 and S2)) .

Page S3 line 5-7 in revised supplementary information:

Mass spectrometry titration (MST) technique was conducted using SHP8400PMS-L online mass spectrometry (Shanghai Sunny Hengping Scientific Instrument Co., Ltd.) with D_2O as the titrator. X-ray diffraction analysis was conducted on a Bruker D8 X-ray analyzer, with a scan rate of 2° per minute, using $\text{Cu K}\alpha$ radiation at 40 kV and 40 mA.

Page S5 line 14-21 in revised supplementary information:

First-principles calculations based on density functional theory (DFT) were conducted in the Vienna *ab initio* simulation package (VASP)^{2,3} with the results dealt by the ALKEMIE platform⁴. The Perdew-Burke-Erzenhof (PBE) functional in the generalized gradient approximation (GGA) is used to describe the commutative correlation term in the DFT calculation.⁵ The cutoff energy was

set as 500 eV for the plane wave basis to ensure the precision of calculations, and the pseudopotential used is projector augmented wave (PAW) pseudopotential.⁶ A vacuum layer of 20 Å was introduced in the calculations to avoid the interaction between images. Both the relaxation convergence for ions and electrons were 1×10^{-5} eV. The k-points of $1 \times 1 \times 1$ were generated with Gamma symmetry automatically. The binding energy (E_b)⁷ was calculated by the equation $E_b = E_{\text{Li/slab}} - E_{\text{slab}} - E_{\text{Li}}$, where $E_{\text{Li/slab}}$ is the total energy of the slab with an adsorbed Li atom, E_{slab} and E_{Li} are the total energies of the slab and a Li atom, respectively. For calculation of reaction free energy, Empirical correction of the Grimme's scheme⁸ (DFT-D3) is used to describe van der Waals interaction. VASPsol program is used to take solvation effect into consideration, and the solvent is water whose dielectric constant is 78.4.⁹ KPOINTS is generated by VASPKIT program¹⁰. The K-Mesh is Monkhorst-Pack Scheme¹¹, and the resolved value is 0.025. The single molecule is in the cube cell with a side length of 25 Å. The Gibbs free energy is obtained by vibration analysis and statistical thermodynamics, and the temperature is 298.15 K.

Page S6 line 1 to Page S9 line 1 in revised supplementary information:

The spontaneous reaction between the main components of passivation layer (Li_2CO_3 , Li_2O , LiOH) and heptafluorobutyric acid (HFA) was first investigated by theoretical calculations. First, the free energy for the reactions between HFA and the main components of the passivation layer (Li_2CO_3 , Li_2O , LiOH) were calculated by VASP5.4.1 program. VASPsol program is used to take solvation effect into consideration, and the solvent is water whose dielectric constant is 78.4.³⁵ KPOINTS is generated by VASPKIT program³⁶. The K-Mesh is Monkhorst-Pack Scheme³⁷, and the resolved value is 0.025. The single molecule is in the cube cell with a side length of 25 Å. The Gibbs free energy is obtained by vibration analysis and statistical thermodynamics, and the temperature is 298.15 K. The calculated reaction equation and the calculated ΔG_r are listed below.

$$\Delta G_r = -197.068 \text{ kJ mol}^{-1}$$

$$\Delta G_r = -415.621 \text{ kJ mol}^{-1}$$

$$\Delta G_r = -278.756 \text{ kJ mol}^{-1}$$

The results of the calculated reaction free energy showed that the reaction of HFA with Li_2CO_3 , Li_2O , and $LiOH$ are thermodynamically spontaneous.

Figure. S1 XRD patterns of $LiOH$, Li_2CO_3 and Li_2O before and after reaction with excess HFA.

After the theoretical proof, experimental verification of the above reaction equation was carried out. XRD (X-ray diffraction) was employed to characterize the changes of reactants before and after the reaction. According to the reaction equations, Li_2CO_3 , Li_2O , $LiOH$ were first reacted with excess HFA, and the reacted products were dried to remove H_2O and any residual HFA. The XRD patterns of $LiOH$, Li_2CO_3 and Li_2O before and after reaction with excess HFA are shown in new Figure S1. The characteristic peaks of $LiOH$, Li_2CO_3 and Li_2O all disappeared completely after the reaction, and the characteristic peaks of the XRD patterns of all reaction products were consistent and identical, indicating the production of HFA-Li

Figure. S2 The TOF-SIMS depth sputter curves of Li surface before and after HFA treatment

TOF-SIMS was used to detect changes in the components of the Li surface before and after HFA treatment. The new Figure. S2a shows the components of the Li surface with sputtering time before HFA treatment. The signal of LiO_2H_2^- and LiO_2H^- can be attributed to the contribution from LiOH. And the strong signals of LiO_2H_2^- and LiO_2H^- suggest that the component of the passivation layer on Li surface is dominated by LiOH. An obvious signal of LiO^- can also be observed. The signal of LiO^- is from LiOH or Li_2O . And the signal of LiCO_3^- usually comes from Li_2CO_3 . The above results show that lithium is covered with a passivation layer composed of Li_2CO_3 , LiOH, and Li_2O , and LiOH is the main component of the passivation layer. And there is also a small amount of Li_2O that is mixed in with the lithium metal. This result is consistent with the previous work.³⁸

After HFA treatment, the signal coming from the component of LiOH has been significantly reduced (Figure. S2b). And the signal of LiO^- can be attributed to the contribution of Li_2O and lithium heptafluorobutyrate. More importantly, a series of ionic fragments appear corresponding to the formation of lithium heptafluorobutyrate. The above results directly demonstrate that HFA pretreatment can successfully remove the surface passivation layer and construct a protective lithium fluorinatedcarboxylate interface on the Li surface. (It should be noted that TOF-SIMS can only be used to detect changes in components and cannot be used to compare the thickness of different coatings. This is because TOF-SIMS analysis has a selected sputtering property, i.e., the sputtering rate is different for different component surfaces. Typically, TOF-SIMS has a faster etch rate for organic materials than for inorganic materials.)

We would like to thank the referee again for taking the time to review our manuscript.

References

1. von Aspern N, Roschenthaler GV, Winter M, Cekic-Laskovic I. Fluorine and Lithium: Ideal Partners for High-Performance Rechargeable Battery Electrolytes. *Angew Chem Int Ed Engl* **58**, 15978-16000 (2019).
2. Meng YS, Srinivasan V, Xu K. Designing better electrolytes. *Science* **378**, eabq3750 (2022).
3. Yu Z, *et al.* Rational solvent molecule tuning for high-performance lithium metal battery electrolytes. *Nature Energy*, (2022).
4. Niu C, *et al.* Balancing interfacial reactions to achieve long cycle life in high-energy lithium metal batteries. *Nature Energy* **6**, 723-732 (2021).
5. Fan X, *et al.* Non-flammable electrolyte enables Li-metal batteries with aggressive cathode chemistries. *Nat Nanotechnol* **13**, 715-722 (2018).
6. Zheng J, *et al.* Electrolyte additive enabled fast charging and stable cycling lithium metal batteries. *Nature Energy* **2**, (2017).
7. Liu K, *et al.* Lithium Metal Anodes with an Adaptive "Solid-Liquid" Interfacial Protective Layer. *J Am Chem Soc* **139**, 4815-4820 (2017).
8. Kim MS, *et al.* Langmuir - Blodgett artificial solid-electrolyte interphases for practical lithium metal batteries. *Nature Energy* **3**, 889-898 (2018).
9. Zhu B, *et al.* Poly(dimethylsiloxane) Thin Film as a Stable Interfacial Layer for High-Performance Lithium-Metal Battery Anodes. *Adv Mater* **29**, (2017).
10. Liu Y, *et al.* An Artificial Solid Electrolyte Interphase with High Li-Ion Conductivity, Mechanical Strength, and Flexibility for Stable Lithium Metal Anodes. *Adv Mater* **29**, (2017).
11. Li NW, Yin YX, Yang CP, Guo YG. An Artificial Solid Electrolyte Interphase Layer for Stable Lithium Metal Anodes. *Adv Mater* **28**, 1853-1858 (2016).
12. Chen H, *et al.* Uniform High Ionic Conducting Lithium Sulfide Protection Layer for Stable Lithium Metal Anode. *Advanced Energy Materials* **9**, (2019).
13. Gao Y, *et al.* Polymer-inorganic solid-electrolyte interphase for stable lithium metal batteries under lean electrolyte conditions. *Nat Mater* **18**, 384-389 (2019).
14. Huang Z, Choudhury S, Gong H, Cui Y, Bao Z. A Cation-Tethered Flowable Polymeric Interface for Enabling Stable Deposition of Metallic Lithium. *J Am Chem Soc* **142**, 21393-21403 (2020).
15. Yu Z, *et al.* A Solution - Processable High - Modulus Crystalline Artificial Solid Electrolyte Interphase for Practical Lithium Metal Batteries. *Advanced Energy Materials* **12**, (2022).
16. Xu K. "Charge-Transfer" Process at Graphite/Electrolyte Interface and the Solvation Sheath Structure of Li^[sup +] in Nonaqueous Electrolytes. *Journal of The Electrochemical Society* **154**, (2007).
17. Zhang XQ, *et al.* Highly Stable Lithium Metal Batteries Enabled by Regulating the Solvation of Lithium Ions in Nonaqueous Electrolytes. *Angew Chem Int Ed Engl* **57**, 5301-5305 (2018).
18. Xie Y-X, *et al.* Succinic anhydride as a deposition-regulating additive for dendrite-free lithium metal anodes. *Journal of Materials Chemistry A* **9**, 17317-17326 (2021).
19. Peng Z, *et al.* Enhanced Stability of Li Metal Anodes by Synergetic Control of Nucleation and the Solid Electrolyte Interphase. *Advanced Energy Materials* **9**, (2019).
20. Chen L, Fan X, Ji X, Chen J, Hou S, Wang C. High-Energy Li Metal Battery with Lithiated Host. *Joule* **3**, 732-744 (2019).
21. Meisner QJ, *et al.* Lithium-sulfur battery with partially fluorinated ether electrolytes: Interplay between capacity, coulombic efficiency and Li anode protection. *Journal of Power Sources* **438**,

- (2019).
22. Su T-T, *et al.* Heteroatom-rich polymers as a protective film to control lithium growth for high-performance lithium-metal batteries. *Journal of Power Sources* **521**, (2022).
 23. Kang D, *et al.* In-situ organic SEI layer for dendrite-free lithium metal anode. *Energy Storage Materials* **27**, 69-77 (2020).
 24. Yang Q, Hu J, Meng J, Li C. C-F-rich oil drop as a non-expendable fluid interface modifier with low surface energy to stabilize a Li metal anode. *Energy & Environmental Science* **14**, 3621-3631 (2021).
 25. Zhang W, Zhuang Houlong L, Fan L, Gao L, Lu Y. A “cation-anion regulation” synergistic anode host for dendrite-free lithium metal batteries. *Science Advances* **4**, eaar4410.
 26. Ma J, Liu M, He Y, Zhang J. Iodine Redox Chemistry in Rechargeable Batteries. *Angew Chem Int Ed Engl* **60**, 12636-12647 (2021).
 27. Xing M, Zhao ZZ, Zhang YJ, Zhao JW, Cui GL, Dai JH. Advances and issues in developing metal-iodine batteries. *Materials Today Energy* **18**, (2020).
 28. Jia W, Wang Q, Yang J, Fan C, Wang L, Li J. Pretreatment of Lithium Surface by Using Iodic Acid (HIO₃) To Improve Its Anode Performance in Lithium Batteries. *ACS Appl Mater Interfaces* **9**, 7068-7074 (2017).
 29. Dong J, Dai H, Fan Q, Lai C, Zhang S. Grain refining mechanisms: Initial levelling stage during nucleation for high-stability lithium anodes. *Nano Energy* **66**, (2019).
 30. Guo W, Zhang W, Si Y, Wang D, Fu Y, Manthiram A. Artificial dual solid-electrolyte interfaces based on in situ organothiol transformation in lithium sulfur battery. *Nat Commun* **12**, 3031 (2021).
 31. Yoo D-J, Kim KJ, Choi JW. The Synergistic Effect of Cation and Anion of an Ionic Liquid Additive for Lithium Metal Anodes. *Advanced Energy Materials* **8**, (2018).
 32. Ye H, *et al.* Synergism of Al-containing solid electrolyte interphase layer and Al-based colloidal particles for stable lithium anode. *Nano Energy* **36**, 411-417 (2017).
 33. Tan SJ, *et al.* Nitriding-Interface-Regulated Lithium Plating Enables Flame-Retardant Electrolytes for High-Voltage Lithium Metal Batteries. *Angew Chem Int Ed Engl* **58**, 7802-7807 (2019).
 34. Zheng G, *et al.* Additives synergy for stable interface formation on rechargeable lithium metal anodes. *Energy Storage Materials* **29**, 377-385 (2020).
 35. Mathew K, Sundararaman R, Letchworth-Weaver K, Arias TA, Hennig RG. Implicit solvation model for density-functional study of nanocrystal surfaces and reaction pathways. *J Chem Phys* **140**, (2014).
 36. Wang V, Xu N, Liu J-C, Tang G, Geng W-T. VASPKIT: A user-friendly interface facilitating high-throughput computing and analysis using VASP code. *Computer Physics Communications* **267**, (2021).
 37. Monkhorst HJ, Pack JD. Special points for Brillouin-zone integrations. *Physical Review B* **13**, 5188–5192 (1976).
 38. Otto S-K, *et al.* Storage of Lithium Metal: The Role of the Native Passivation Layer for the Anode Interface Resistance in Solid State Batteries. *ACS Applied Energy Materials*, (2021).

REVIEWERS' COMMENTS

Reviewer #2 (Remarks to the Author):

The authors have well addressed the previous concerns, and the manuscript is recommended for publishing on Nat Commun.

Reviewer #3 (Remarks to the Author):

The revised manuscript has been significantly improved and all previous concern regarding the data and data analysis has been addressed. Nevertheless, the level of novelty does not seem to meet the standards of the journal. Indeed the concept that is presented can be seen as a variation of the work published by W. Jia et al., which is now cited as ref. 10 in the manuscript. In comparison to the latter work, a possible improvement would be the suppression of the shuttling of iodine-based specie. However, there is no clear evidence that the approach presented in ref. 10 is plagued by such shuttling effect. It is assumed by the authors on the basis of a parasitic effect occurring in metal-iodine systems, as described in two recent reviews (refs. 11 and 12); but metal-iodine batteries are a very different system in which iodine is the electroactive specie at cathode. As described in ref. 11, "both iodine and polyiodide species formed during charge/discharge cycling easily dissolve into electrolytes and then uncontrollably diffuse toward the anode side." Since in the case of an anode pre-treatment, LiI would be readily present at the anode side and embedded within the SEI, there is no clear evidence this brings an issue at all.

REVIEWERS' COMMENTS

Reviewer #2 (Remarks to the Author):

The authors have well addressed the previous concerns, and the manuscript is recommended for publishing on Nat Commun.

Response: Thank you very much for your positive comments!

Reviewer #3 (Remarks to the Author):

The revised manuscript has been significantly improved and all previous concern regarding the data and data analysis has been addressed. Nevertheless, the level of novelty does not seem to meet the standards of the journal. Indeed the concept that is presented can be seen as a variation of the work published by W. Jia et al., which is now cited as ref. 10 in the manuscript. In comparison to the latter work, a possible improvement would be the suppression of the shuttling of iodine-based specie. However, there is no clear evidence that the approach presented in ref. 10 is plagued by such shuttling effect. It is assumed by the authors on the basis of a parasitic effect occurring in metal-iodine systems, as described in two recent reviews (refs. 11 and 12); but metal-iodine batteries are a very different system in which iodine is the electroactive specie at cathode. As described in ref. 11, "both iodine and polyiodide species formed during charge/discharge cycling easily dissolve into electrolytes and then uncontrollably diffuse toward the anode side." Since in the case of an anode pre-treatment, LiI would be readily present at the anode side and embedded within the SEI, there is no clear evidence this brings an issue at all.

Response: Thank you very much for your positive comments. Regarding the reviewer's concerns about the validity of our proposed I^3^-/I^- shuttle effect, we would like to refer to a recent paper published in Nature Energy (Rejuvenating dead lithium supply in lithium metal anodes by iodine redox. Nature Energy 6, 378-387 (2021)). This paper provides evidence that the I^3^-/I^- shuttle effect is indeed present in lithium-ion battery systems, as we had hypothesised. The authors achieved longer battery life in LiFePO₄ batteries by introducing the redox reaction of iodide ions. However, this approach is not compatible with high voltage cathode materials such as NMC811. Therefore, the iodic pre-treatment strategy is not suitable for high voltage cathode materials and the LiI protective layer is difficult to maintain during cycling. In contrast, our proposed HFA pre-treatment has been successfully applied to 50- μ m-thin Li||high-loading-NMC811 full batteries to achieve high cycle stability. We hope this clarification addresses the reviewer's concerns. Thank you again for your valuable feedback.